



# Escarpment retreat rates derived from detrital cosmogenic nuclide concentrations

Yanyan Wang[1], Sean D. Willett[1]

[1]Department of Earth Sciences, ETH Zürich, Zürich, 8092, Switzerland

*Correspondence to*:  Yanyan Wang (yanyan.wang@erdw.ethz.ch)

**Abstract.** High-relief great escarpments at passive margins present a paradoxical combination of high relief topography, but low erosion rates suggesting low rates of landscape change. However, vertical erosion rates do not offer a straightforward metric of horizontal escarpment retreat rates, so we attempt to address this problem in this paper. We show that detrital cosmogenic nuclide concentrations can be interpreted as a directionally-dependent mass flux to characterize patterns of non-

vertical landscape evolution, e.g. an escarpment characterized by horizontal retreat. We present two methods for converting cosmogenic nuclide concentrations into escarpment retreat rates and calculate the retreat rates of escarpments with published cosmogenic [10]Be concentrations from the western Ghats of India. Escarpment retreat rates of the Western Ghats inferred from this study vary within a range of 100s m/Ma to 1000s m/Ma. We show that the current position and morphology of the Western Ghats are consistent with an escarpment retreating at a near constant rate from the coastline since rifting.

## 1 Introduction

Passive continental margins exhibit a characteristic morphology with a high-relief escarpment separating a low-relief inland high plateau and the low, flat coastal plain (Fig. 1). The edge of an escarpment often coincides with a major, even a continental, water divide. The  escarpment that separates the plateau and the coast plain often exhibits local relief of over 1km, even reaching heights  exceeding 2 km. These great escarpments extend hundreds of kilometres parallel to the coast

along rift margins and are typically found 30-200 km inland from the coastline (Linari et al., 2017; Persano et al., 2002). Examples of passive margin escarpments and their age of formation include the Red Sea margin (10-5 Ma), the Western Ghats in India (84 Ma) (Eagles and Hoang, 2014), the Serra do Mar escarpment in Brazil (125 Ma), the Drakensberg escarpment (130 Ma) in South Africa, the Queensland escarpment in Australia (150 Ma) and Blue Ridge escarpment in the US (200Ma) (Matmon et al., 2002).

The absence of active tectonics at old rift margins makes the formation and persistence of escarpments a long-debated problem. One major dispute is whether an escarpment is geomorphologically static or dynamic in the sense of rates of change of erosion or back-cutting or retreat of the escarpment away from the coast. Most researchers agree that an escarpment originates from rifting-related processes, forming at the edge of a rift graben, and subsequently migrating inland



to their modern position (Sacek et al., 2012; Tucker and Slingerland, 1994; Gilchrist and Summerfield, 1990), but it is still debated as to whether this happens as a continuous process or occurs rapidly following rifting with subsequent slowing or stalling, so that the modern geomorphic feature is static (Beauvais et al., 2016; Bonnet et al., 2016; Beauvais et al., 2008). Relatively static escarpments are supported by low denudation rates. Apatite fission track ages (AFT) and (U-Th)/He ages on the escarpment-side coastal plain are rarely reported to be significantly younger than the break-up age (Persano et al., 2006;

Persano et al., 2002; Cockburn et al., 2000), suggesting that escarpments form and retreat rapidly following break-up, but subsequently slow. Erosion rates from in-situ cosmogenic nuclides concentrations of escarpment-draining basins are also very low, with rates on the order of 10s of meters per million years (Portenga and Bierman, 2011), supporting the idea that older escarpments are gynomorphically static.

Alternatively, numerical studies of escarpment topography evolution suggest a much more dynamic and long-lived geomorphic evolution (Braun, 2018; Willett et al., 2018; Sacek et al., 2012; van der Beek et al., 2002; Tucker and Slingerland, 1994; Kooi and Beaumont, 1994). Braun (2018) presented a parameterization of erosion and retreat of an escarpment based on fluvial erosion and diffusion, and showed that this would lead to a constant rate of retreat over time. Willett et al. (2018) also argued that escarpment processes should evolve to maintain a constant form of an escarpment with

a constant rate of backward retreat where escarpment slope maintains a balance with rock advection driven by retreat. Given a constant rate of escarpment retreat and formation during rifting, a retreat rate of order ~1 km/Ma is suggested from the distance of most escarpments from the coastline (Seidl et al., 1996; Oilier, 1982).

Testing of these models is difficult in that measuring a horizontal retreat rate of a geomorphic feature is quite difficult. Direct evidence for escarpment retreat comes from terrace deposits found atop of the Blue Ridge escarpment crest and beheaded

drainages on the plateau side of the escarpment (Prince et al., 2010). Erosion rates are easier to measure and have been estimated from sediment budgets measured in the offshore (Campanile et al., 2008) and from concentrations of cosmogenic radionuclides (e.g. de Souza et al., 2019; Linari et al., 2017 ; Salgado et al., 2014). DCN [10]Be-derived erosion rates used to calculate retreat rates of the South Eastern Australian escarpment, yielded rates of 40-80 mm/ka over the last few 100s of thousand years (Godard et al., 2019).

The lack of post-rift thermochronometric cooling ages and the very low rates of erosion derived from [10]Be concentrations suggest little geomorphic modification of the landscape and only slow retreat of the escarpment. Given that the erosion associated with the relief of even the largest escarpments is under 2 kilometers, it is likely to be too small to be measured by thermochronometry except at very high geothermal gradients. It is likely that cooling ages largely reflect diffusive or exhumational cooling associated with lithospheric cooling following rifting and are insensitive to the subsequent erosion

associated with escarpment retreat. Cosmogenic isotope concentrations also have a problem, in that escarpment retreat



Earth **Surface**
Dynamics
Discussions



suggests a spatially variable erosion rate so that catchment wide averages, even when correctly measuring mean rates, may not be representative of the local process rates responsible for escarpment retreat.

In this paper, we present a new, systematic method for interpreting detrital cosmogenic isostope concentrations in terms of horizontal retreat rates of an escarpment. The method is based on the physical principle of the models of Braun (2018) and
Willett et al. (2018) that argued that an escarpment should evolve into a morphology that drives horizontal retreat at a constant rate. We demonstrate that these conditions are exhibited by the Western Ghats escarpment in India, which shows channel profiles consistent with the concept of a steady, retreating escarpment with occasional river capture from the upper plateau. Under conditions of steady horizontal escarpment retreat, we demonstrate that in-situ detrital cosmogenic nuclides concentrations can be interpreted directly in terms of an average horizontal retreat rate of a catchment. We present two
methods for the calculation of horizontal retreat rates and demonstrate these methods using published detrital $^{10}$Be concentrations from the Western Ghats in India.

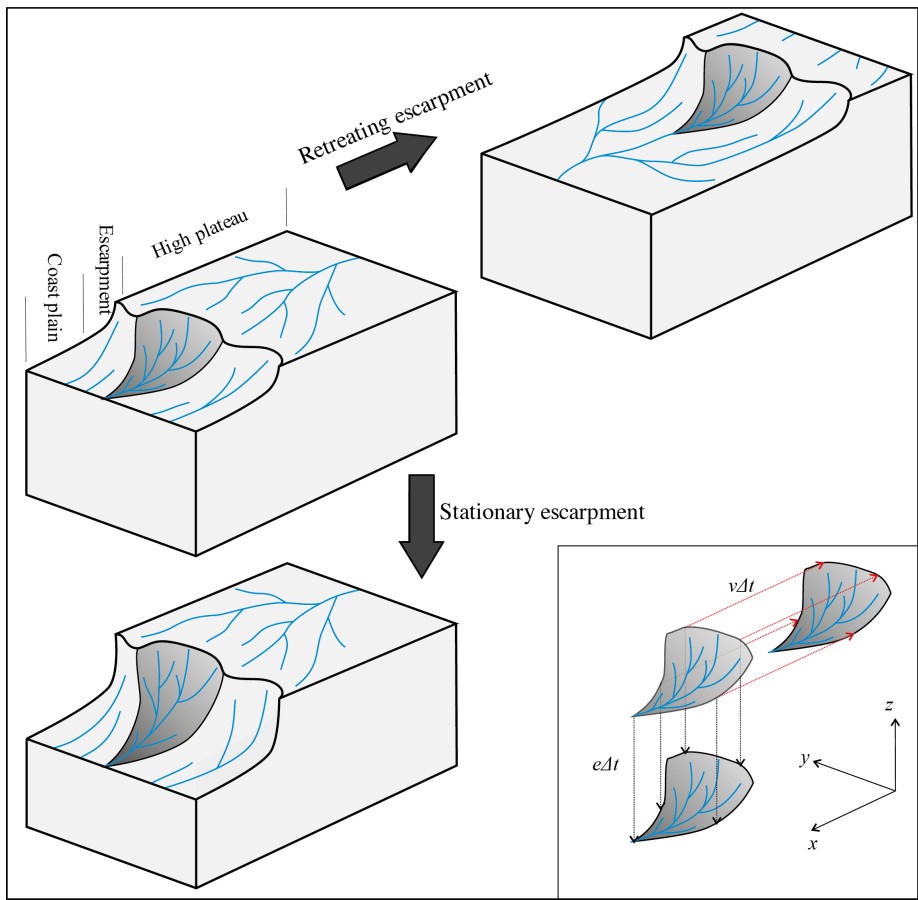

**Figure 1: Conceptual diagram showing two scenarios of escarpment evolution: a retreating escarpment or a down-cutting**
**escarpment with stationary water divide. The grey surface denotes the surface of an elemental escarpment-draining catchment.**



**Inset is the mathematical representative of the two scenarios as motion relative to the rock. $\Delta t$ denotes the unit time, $v$ denotes retreat rate, $e$ denotes a vertical erosion rate, red arrows indicate the horizontal retreat vector and black arrows indicate the vertical erosion vector.**

## 2 Model for Escarpment Retreat

### 2.1 Concept of escarpment retreat

High-relief escarpments along rifted continental margins pose a stark morphologic contrast with their neighboring low-relief plateau and coastal plain. The typical dimensions of an escarpment with 5 to 20 kms of extent normal to the margin imply enough drainage area that a well-developed river network is present and dominates the geomorphic processes. Normal scaling relationships between slope and area predict high normalized channel steepness for most escarpments, so the

observed rates of erosion, which are low, are surprising.

To maintain generality, one can consider two evolution scenarios in terms of river incision: downcutting of the topography with a stationary escarpment (Gunnell and Fleitout, 1998) and back-cutting or retreat of the escarpment with migration of the water divide (Tucker and Slingerland, 1994) (Figure 1). The downcutting model can be driven by base level fall or it can

involve a change in relief with a fixed base level and a stationary water divide and escarpment. In the stationary escarpment and water divide scenario, the position of an elemental catchment remains stationary. Assuming that erosion rates on the plateau are negligible, the surface of the catchment will downcut only if the escarpment front becomes steeper and shorter. Without a change in the coastal elevation, erosion is focussed on the escarpment. In the retreating escarpment scenario, the escarpment front retreats towards the inland plateau. Headward erosion of escarpment rivers drives retreat of the escarpment,

widening the coastal plain and enlarging the escarpment draining basins, although the overall height and morphology of the escarpment remains constant, neglecting the increase in elevation of the base of the escarpment as the coastal plain increases in length (Willett et al., 2018).

These models can be described in terms of a surface moving in either a vertical or horizontal direction with respect to its

underlying rock (Fig. 1). Although, as argued by Gunnell and Harbor (2010), the geometry of an escarpment cannot remain strictly self-similar or uniform during its evolution at geological timescales, we assume that morphologic changes are small and an instantaneous erosion or retreat velocity is characteristic of the average change over longer timescales. For vertical erosion, this is essentially the assumption made in treating cosmogenic isotope concentrations, converting concentration to catchment average erosion rate. Here we propose that the retreat velocity should be treated in the same manner, representing

it as a horizontal motion of the catchment surface. In this case, the change of the surface can be characterised by a vector in which the magnitude represents the retreat velocity and the direction represents the retreat direction, taken with respect to the solid earth. In this paper, we will investigate the implications of these end-member models for erosional fluxes.



## 2.2 Southern Western Ghats

### 2.2.1 Geological and morphological features

The escarpment on the west margin of India is a well-recognised escarpment. It extends parallel to the coast for 1500 km and defines the mountainous region of the Western Ghats (Fig. 2). The western margin of India rifted from Madagascar at ~84 Ma (Eagles and Hoang, 2014), with a secondary rifting from the Seychelles affecting the northern segment of the Indian margin (Torsvik et al., 2013). The Western Ghats exhibit relief of 1000 m to 2600 m. The escarpment is heterogeneous in

terms of bedrock geology and morphology from north to south. The northern Western Ghats (21˚N – 16˚N) lies on the Deccan igneous province (the Deccan Traps), whilst the southern Western Ghats (16˚N – 10˚N) is located in the Archean-Proterozoic metamorphic shield. The sinuosity of the escarpment divide varies, with a higher sinuosity of 2.73 in the northern Western Ghats and a value of 2.2 for the southern Western Ghats (Matmon et al., 2002). The southern Western Ghats escarpment (abbr. as SWG escarpment hereafter) is 30 km to 90 km from the coast line (Fig. 2). The SWG escarpment

usually coincides with the continental water divide, but in some areas, the water divide is located inland from the morphologic escarpment. Although the history of the river topologic structure is not known, most of the morphologically flat regions currently draining to the west are small in area and are consistent with relatively recent capture of these drainages from east-directed to west-directed. Consequently, some escarpment-draining basins may have gained drainage area from the plateau, and we distinguish between rivers that have a headwater divide on top of the escarpment from those that include

drainage area from the plateau (e.g. basin A and basin B in Fig. 2).

Escarpment rivers in the SWG are bedrock rivers cutting into the Precambrian metamorphic basement. The morphology of the rivers draining the escarpment differ primarily due to their initiation on the escarpment or landward of the escarpment on the plateau (Fig. 3). Rivers initiating on the escarpment are characterised by a long, low-slope reach on the coastal plain and

abrupt steepening at the escarpment front (Fig. 3a). This is particularly evident in transformed χ-elevation river profiles, which normalize the river profiles for drainage area (Perron and Royden, 2013). A typical χ-elevation profile of these escarpment front-initiated rivers is composed of two near-linear segments: the coastal plain reach and the short and steeper escarpment-draining reach (Fig. 3b). This characteristic χ-elevation profile indicates the transient state of the escarpment topography, and is consistent with the model of a moving escarpment with all erosion focused on the escarpment face

(Willett et al., 2018). For plateau-initiated rivers, the channel profile and χ profile have an additional low-slope 'tail' at low drainage area, representing the reach on the plateau (Fig. 3c, 3d).





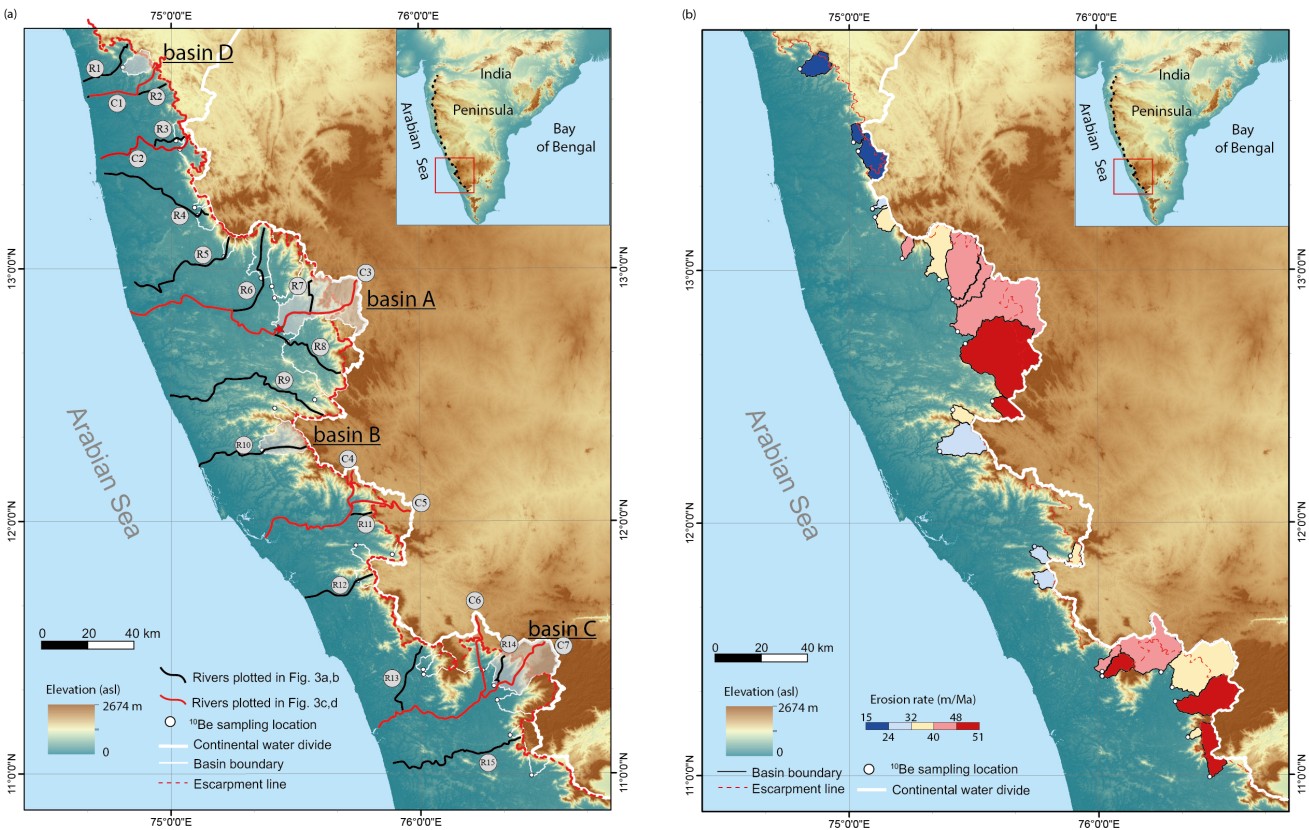

**Figure 2 (a) Topographic map of the southern Western Ghats escarpment in India (location is indicated in the inset figure). The black dashed line denotes the escarpment defining the Western Ghats, and we take the southern limit of the Deccan Traps as the boundary between the northern and southern segments. The $^{10}$Be sample locations of Mandal et al. (2015) are indicated by grey circles. (b) Cosmogenic $^{10}$Be-derived erosion rates. Erosion rate is recalculated from the published concentrations following the method of Lupker et al. (2012).**





**Figure 3 River profiles and corresponding transformed χ profiles in the south Western Ghats. Locations of these rivers are indicated in Fig. 2a. Rivers are extracted from 90 m SRTM data and use a threshold drainage area of 1 km². The χ value of rivers is calculated using a base level of sea level, concavity of 0.42 and precipitation is not included. (a-b) Profiles of escarpment rivers that initiate from the escarpment. (c-d) Profiles of escarpment rivers that initiate from the plateau interior and drain through the escarpment.**

### 2.2.2 Methods of river profile analysis

In order to calculate a scaled river profile, it is necessary to assume or estimate the concavity of the profile (Perron and Royden, 2013). We evaluated the slope-area scaling of escarpment-draining rivers (Figure 4). The channel slope and drainage area data were extracted with the MATLAB-based software TopoToolBox 2 (Schwanghart and Scherler, 2014). We calculated the average slope and drainage area over predefined river segments. River segments were defined with a length of





1 km but break at confluences and were limited by both a threshold slope and drainage area. Recognizing that there were two sets of data, corresponding to the escarpment and the coastal plain, we searched for an optimal break point in slope-area space, searching within the red-dashed line box in Fig. 4b.

We found concavities of 0.3 to 0.6 for the SWG rivers, which is typical for bedrock rivers (Snyder et al., 2000). We used a

mean value of 0.42 for the concavity, in order to calculate the normalized steepness index for each of the major escarpment rivers. Conventionally, normalized steepness index is taken as a proxy for erosion rate (Kirby and Whipple, 2012). However, for an escarpment, uplift rate is likely to be limited to the isostatic response to erosion, and the erosion rate should be reflective rather of the erosion associated with the escarpment retreat. Willett et al. (2018) analysed this problem and demonstrated that the slope-area scaling for a river retreating in a direction opposite to its flow should scale according to:

$$Sl = -\left(\frac{v}{K}\right)^{\frac{1}{n-1}} A_d^{-\frac{m}{n-1}} \qquad n > 1, \tag{1}$$

where $v$ is the retreat rate, $Sl$ is the local channel slope, $A_d$ is the upstream drainage area and $K$ is the erodibility constant, $m$ and $n$ are positive empirical constants. The steepness of a channel following this scaling would be:

$$k_s = \left(\frac{v}{K}\right)^{\frac{1}{n-1}} \qquad n > 1, \tag{2}$$

This relationship implies a lower concavity (m/n-1) than rivers in equilibrium with vertical uplift, so it is interesting that the concavities we find are close to global averages. This suggests that the assumptions made by Willett et al. (2018) of a steady, 1-D river normal to the escarpment with continuous area gain at the channel head might not be appropriate. Sinuous, branching rivers in a transient state due to discrete area capture might fit such a model on average, but not for individual rivers. The slope-area relationship (Fig. 4c) also shows the segmented form as in the channel profiles (Fig. 4b).

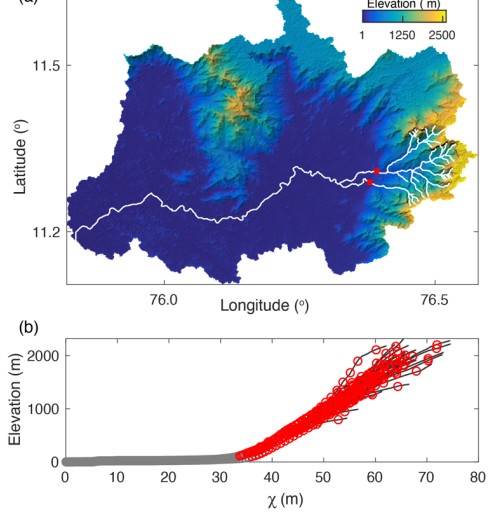

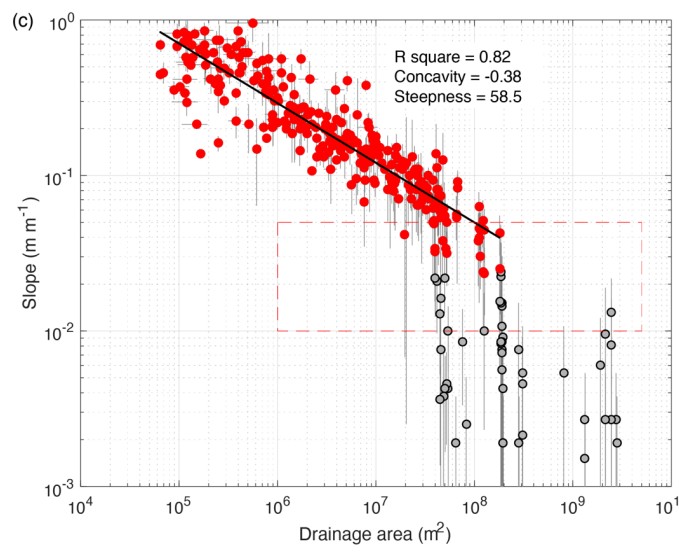






**Figure 4 (a) The Chaliyar River catchment in Western Ghats, India. Red square indicates potential transition point from coastal plain to escarpment channel. Outlet of the drainage basin is at the coast of the Arabian Sea. Channels are extracted from the DEM using a threshold drainage area of 0.5 km² (approximately 6 DEM grid cells). (b) The Elevation-χ profiles of channels shown in (a). Grey circles indicate coastal plain reaches; open red circles are escarpment reaches and are used in slope-area regression. χ is**
**calculated using a concavity of 0.38. (c) Channel slope-drainage area plot in log space. Uncertainty is one sigma. The box marked in the red-dashed line is the range for searching of the threshold point between coastal plain and escarpment (see methods).**

### 2.2.3 Escarpment retreat from river profile analysis

The segmented form of the escarpment-draining rivers is consistent with models of escarpment retreat with a lower reach on the coastal plain, where the gradient is sufficient to transport eroded sediment, but is not incising bedrock. On the upper
reach, incision rates are high, but have a pattern that results in horizontal retreat of the escarpment as well as the drainage divide. The normalized steepness indices derived from slope-drainage area plots or from the normalized channel profiles show a constant value for the escarpment reaches, consistent with a constant rate of erosion, but also consistent with a constant horizontal retreat rate (Willett et al., 2018). Furthermore, river profiles have the same form, but the lengths of the various reaches are highly variable, even scaled into χ space. This suggests that the kinked profile form is not the result of a
temporal change in uplift rate common to all rivers, in which case the χ scaling would collapse the profiles onto a common form. Rather they are consistent with an escarpment retreat model in which the lower reach is graded to a low slope sufficient to transport sediment from the eroding escarpment reach, and the steep segment is adjusted to erode the escarpment (Willett et al., 2018).

River that include plateau reaches (Figure 3c,d) are scattered throughout the study area, intermixed with the escarpment rivers. This suggests that they are not the response of temporal variations in uplift rate, i.e. are not moving knickpoints in response to baselevel changes, or they would be clustered together spatially and have common chi profiles, at least within single drainage basins. Rather they appear to be the response to capture of river reaches from the plateau to the coastal plain (Giachetta and Willett, 2018).

The values of the normalized steepness on the escarpment reaches are relatively high compared to other rivers globally, but particularly for the observed erosion rates (Fig. 5). In fact, the values of channel steepness from the Western Ghats are amongst the highest in the world at the observed erosion rates. Although the bedrock is relatively erosion resistant, rainfall is also relatively high, so there is no obvious reason for these high values in a region where the tectonic uplift rates are likely to
be low and not localized to the escarpment.

Taken together, these observations suggest that the Ghats escarpment is actively retreating to the east. The high relief is likely to be old and inherited, rather than the result of recent uplift, and erosion is focused on the escarpment, driving the escarpment horizontally, rather than eroding the entire landscape downward. This suggests that the [10]Be concentration data
should be interpreted with this landscape evolution model in mind, and we pursue this in the next section.



Earth **Surface**
**Dynamics**
Discussions

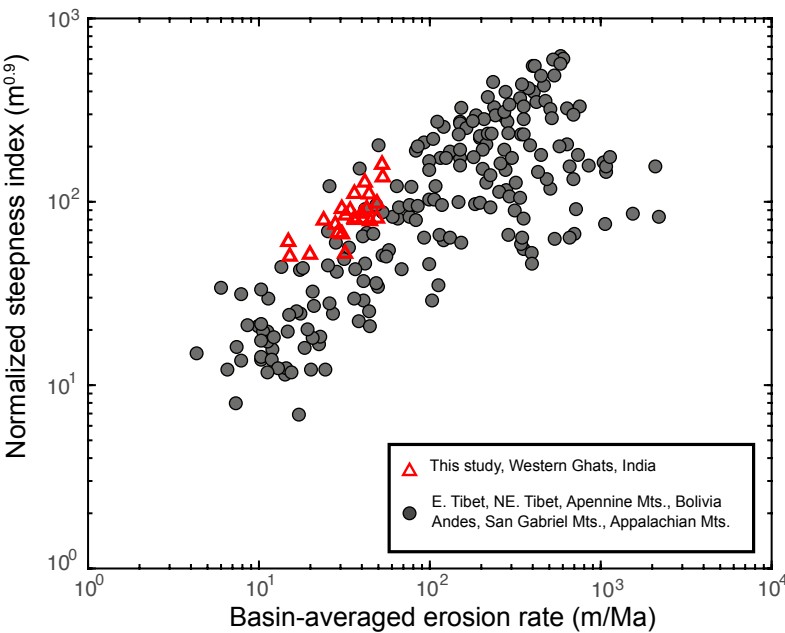

**Figure 5 Normalised channel steepness index and cosmogenic ¹⁰Be-derived basin-averaged erosion rates from the Southern Western Ghats (red-lined triangles) with comparison to a compilation of global data by Kirby and Whipple (2012) (grey dots).**


## 2.3 Erosional flux and rock velocity from cosmogenic isotope concentrations

The use of cosmogenic isotope concentrations to derive conventional erosion rates makes a key assumption of continuous and steady removal of rocks from the earth surface (Lal, 1991). As production of cosmogenic isotope nuclides is fast with respect to erosion rates, steady erosion implies that the concentration profile of a cosmogenic nuclide with depth is also

steady (Niemi et al., 2005). Catchment-wide, detrital cosmogenic-based erosion rates require only a weaker assumption, specifically that equilibrium is maintained between the catchment-integrated quantities of production and removal. The appropriate secular equilibrium state implies that over an appropriate timescale, the number of cosmogenic isotope atoms produced is equal to the number lost by erosion, as integrated over the entire catchment. The total rate of production is constant and is predictable according to the geographic location, topology and exposed lithology of the catchment (Stone,

2000; Lal, 1991). If we assume well-mixed sediment derived from the entire catchment surface, the measured concentration of a cosmogenic nuclide within these sediments is equal to the total catchment production divided by the total volume of eroded rock, or more precisely, the volume of the target mineral bearing the cosmogenic nuclide, e.g. quartz. Production divided by volume can be expressed as the erosional mass flux out of the catchment surface. As a flux, this requires the area of the surface, and in practice this area is calculated from the surface projected onto a horizontal plane, as is done with any





standard digital elevation model. This implicitly defines the erosional flux vector to be vertical. However, in general, the mass flux does not need to be treated as a purely vertical flux. In many geomorphic or tectonic settings such as the escarpment problem described above, the change of the surface is better described with a component in the horizontal direction with respect to the underlying rock, thus defining a flux in a non-vertical direction. At rift escarpments, the mass flux of an escarpment-draining basin can be approximated as purely horizontal and the mass flux is determined by the rate of

escarpment retreat with no vertical component. This suggests a need to redefine the expressions describing erosion rates in terms of $^{10}$Be concentrations, generalizing these for a flux of mass through the Earth's surface in a non-vertical direction.

In the following section, we derive the expression of catchment-wide cosmogenic nuclide production with surface production rate, for arbitrary local rock motion direction. Then we present two methods for calculation of mass flux and velocity from the measured detrital concentration and the catchment-wide nuclide production.

**2.3.1 Catchment average erosion rates from cosmogenic nuclide concentration**

Production of in-situ cosmogenic radionuclides (CRNs) at and near the Earth's surface is a function of the flux of cosmic ray particles at the appropriate energy level (Lal and Chen, 2005). Commonly used in geological studies is the simplified form by approximating the cosmic ray flux into collimated ray flux hitting a target: for a flat surface well exposed to cosmic rays, the production rate of a radionuclide at some physical depth $z$ *(m)* interior to the surface decays exponentially at a constant

rate $\frac{\rho}{\Lambda}$ (cm$^{-1}$) (Lal, 1991):

$$P(x, y, z) = P_0(x, y) exp\left(-\frac{\rho}{\Lambda} z\right) \tag{3}$$

where $\rho$ (gram/cm$^3$) is the density of the target, and $\Lambda$ (gram/cm$^2$) is free path absorption length. A single exponential represents production of nuclide by spallation. For production of nuclides by capture of slow muons and by fast muons, we would need to consider additional depth functions, which are generally taken to be exponential. We include the production of

nuclides due to muons within our calculations, but illustrate production using only spallation for simplicity. The penetration distance is in a general direction, but for incidence on the Earth's surface, $z$ is taken as downward. Volume mass is converted into weight mass with target density $\rho$. A representative volume for the increment of mass $dMass$ (gram) is given by:

$$dMass = \rho dx dy dz, \tag{4}$$

where we assume that flux is in the $z$ direction and $dz$ is large enough that the cosmic ray flux is fully attenuated. The bulk

production of nuclide atoms within the target mass per unit time $M$ (atoms/year) is an integration of the production rate $P(z)$ at depth $z$ through mass:

$$M = \int_0^\infty P(x, y, z) dMass = \int_0^\infty P_0(x, y) exp\left(-\frac{\rho}{\Lambda} z\right) \rho dx dy dz, \tag{5}$$



For the case of a single exponential (Eq. (3)), the bulk production of nuclide atoms $M$ (atoms/year) is obtained by integration over depth, giving:

$M = \Lambda P_0(x,y)dxdy,$         (6)

With surface erosion at a steady rate of $e$, the concentration of a CRN will reach a steady value at both the surface and at depth. This equilibrium will occur with or without considering the radioactive decay of the CRN. At a geographic location $(x,y)$ where the production rate at the earth's surface is $P_0(x,y)$, cosmogenic nuclides build up in the target mineral during the time of exhumation. The concentration of CRN in the mineral grain at the surface is (Granger et al., 2013):

$C_0 = P_0(x,y)\Lambda/(e\rho),$         (7)

where $C_0$ is given as atoms/g in target minerals; $e$ is erosion rate.

 For a detrital measurement of CRN concentration within river sediment at the mouth of a catchment, these quantities must be averaged over the full upstream catchment area. Cosmogenic nuclide concentration of river sediment at the outlet is the basin-averaged concentration and represents the integrated production of CRNs and the integrated erosion rate (Granger et

al., 2013):

$\underline{C} = \underline{P_0}\Lambda/(\underline{e}\rho),$         (8)

where the underbars represent integration over the catchment surface ($S$), which has total area, $A$ (Lupker et al., 2012), so that each quantity is an integral:

$\underline{P_0} = \frac{1}{A}\iint_S P_0(x,y)\,dxdy,$         (9)

$\underline{e} = \frac{1}{A}\iint_S e(x,y)\,dxdy,$         (10)

Estimation of catchment-wide erosion rate is done by solving for $\underline{e}$ in Eq. (8), given a measurement of $\underline{C}$. Complications arise from the calculation of production rates, inclusion of muon production, radioactive decay, and topographic shielding. Muon production was discussed above and involves only adding additional production terms, which are also integrable. The surface production rate can be estimated from scaling relationships for altitude (or atmosphere pressure), geographic location

(latitude and longitude), taking into account production pathway (from neutron spallation, capture of muons and fast muons), or irradiation geometry (Heisinger et al., 2002a; Heisinger et al., 2002b; Masarik et al., 2000; Stone, 2000; Lal, 1991).

Shielding of cosmic rays on an individual surface is a function of the surrounding topography on the skyline, as well as by local slope effects. Cosmic ray shielding generally reduces surface production rate $P_0(x,y)$ but extends the free path

attenuation length by the change of irradiation geometry (Dunne et al., 1999). DiBiase (2018) evaluated the counter-effects of shielding on catchment-wide production rate (via spallation) and attenuation length, although he found that the combined



shielding effect on surface production rate (via spallation) and free path attenuation length is negligible when the valley surface slope is less than 30 degrees (DiBiase, 2018). In general, shielding calculations for detrital CRN data are not very important as most shielding is local, i.e. within the catchment area, and therefore sums to zero during integration.


Radioactive decay is only a factor if erosion rates are low, but can be an issue also with spatially variable erosion rates, as we will discuss later.

This conventional calculation of basin-averaged erosion rates from CRN concentrations is widely used in many geological
studies, and can be done using standard software algorithms (e.g. CRONUS) (Balco et al. 2008) to calculate the basin-averaged erosion rate.

### 2.3.2 Mass Fluxes

Detrital CRN concentrations as expressed above can be thought of as representing a balance between two fluxes. The flux of cosmic rays into a catchment determines the total production in the basin as given by Eq. (9). With a steady state, this
production is balanced by the export of the CRN within eroded sediment. The flux of sediment out of the basin determines the mass through which the CRN is distributed and thus the concentration. Concentration can be expressed in terms of the total production over a time interval divided by the total volume of sediment produced over that time interval:

$$C = \frac{M_c}{V_{rock}} = \frac{\iint_S \Lambda P_0(x,y) dxdy}{\iint_S e(x,y) dxdy}, \tag{11}$$

where $M_c$ is the total production of CRN mass over the catchment with dimensions of moles/time, and $V_{rock}$ is the volume of
rock converted to sediment and exported from the catchment. This has dimensions of volume per time, and can also be expressed as the integral of the erosion rate over the surface of the catchment, $S$. Erosion rate in this context can be regarded as a flux, as it is a volume of rock produced per square area per unit time.

This concept can be generalized if we think of the rock within the Earth as moving at a constant velocity, $F_s$, with respect to
the surface. The flux of rock, $V_{rock}$, through the surface is the scalar (dot) product of the vector $\vec{F_s}$ and the vector normal to the Earth's surface, $S$ :

$$V_{rock} = \vec{F_s} \cdot S, \tag{12}$$

If the vector $F_s$ is vertical, we refer to $F_s$ as erosion rate and the projected area of $S$ as the catchment area, $A$, and we revert to Eq. (8). However, in general, there is no reason for $\vec{F_s}$ to be vertical. In particular, for the problem of escarpment retreat, the
land surface might be better regarded as moving horizontally with respect to the underlying rock (Fig. 1). In this case, we can take $\vec{F_s}$ as horizontal, but the rock flux is still calculated from the scalar product of $F_s$ and the normal to $S$. The general form of Eq. (11) for the rock velocity in any direction becomes:


Earth **Surface** Dynamics
Discussions
$$C = \frac{\iint_S \Lambda P_0(x,y)dxdy}{\vec{F_s}\cdot S},$$ (13)

The concentration of a CRN in sediment is still the ratio of the production of CRN and the dilution into a flux of sediment, but the flux can be generated by a velocity in any direction, and requires that the catchment surface be projected into a plane normal to the direction of motion. For example, Fig. 6 shows a catchment surface together with its projection onto two planes, one horizontal and one vertical. The projection of the surface onto a horizontal plane produces a surface with an area of $A_v$, which we would recognize as the usual catchment area, as calculated from a DEM, for example, noting that this

quantity is also a projection. The projection onto a vertical plane has a different shape and area, denoted as $A_h$. In either case, Eq. (13) holds and we could infer a velocity, $\vec{F_s}$, from a CRN concentration and the appropriately projected area. If we assume the velocity is vertical, $\vec{F_s}$ is equal to the conventional erosion rate, $e$. If we assume that $\vec{F_s}$ is horizontal, we would infer a horizontal retreat rate of the landscape at a velocity, $v$ (Fig. 6).

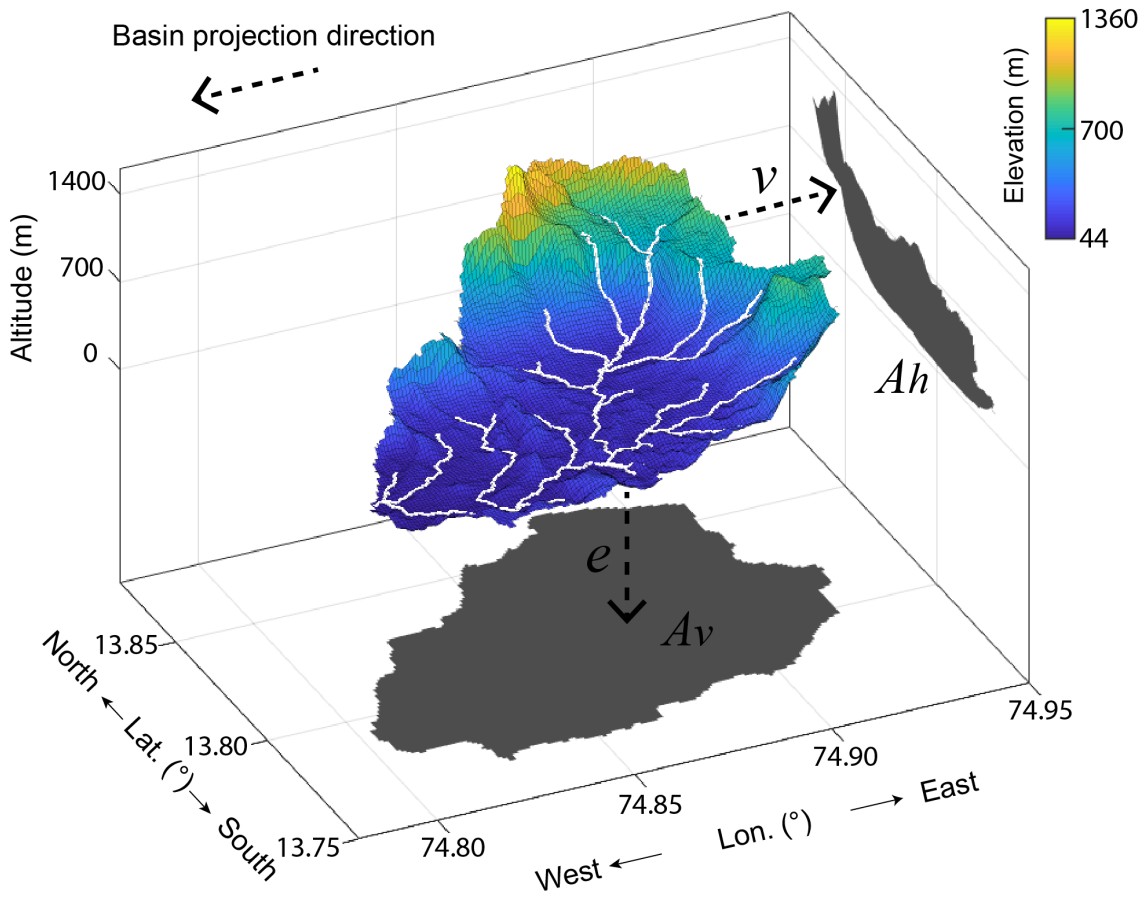


**Figure 6 Depiction of the flux of mass through the surface of a drainage basin from the coastal escarpment of the Western Ghats, India. Basin position is indicated as basin D in Fig. 2. Spatial surface of drainage basin and its projection in the vertical and a**





**horizontal direction. Projected areas are $A_v$ and $A_h$, respectively. Thick white lines denote channels extracted from the DEM for a minimum drainage area of 1 km².**


It is important to note that changing the assumed direction of the rock with respect to the surface does not change any of the basic physical processes. Production of a CRN is unaffected as it is still produced within the same depth range, $\Lambda$, and is determined exclusively by the area, elevation and geometry of the catchment. Provided the assumption of steady state holds, the total production over the catchment is equal to total export, and this calculation is independent of the direction of rock

motion or the path of integration. A concentration measurement is thus constraining the flux of rock, and it is only convention that regards this flux as characteristic of a downward motion of the surface.

Escarpment retreat or other non-vertical motion of the earth's surface can be regarded as being simply a product of spatially variable surface lowering. To illustrate this point, consider the example shown in Fig. 7. A land surface is represented by a

one-dimensional profile represented by an exponential function. This surface is back-cutting or retreating to the right, maintaining its form. That motion can be represented by a vertical erosion rate that is variable in space (Fig. 7b), or by a horizontal back-cutting at a constant rate, $v$, (Fig. 7c). Each mathematical description results in the same change in the surface of the Earth with time, the same flux of rock (Appendix 1), the same catchment integrated concentration of detrital CRNs. The physical erosion processes would be the same; rock is eroded and removed by gravity processes which operate

downslope, and the path through the production zone would have the same vertical component with respect to the surface, and so would produce the same surface concentration (Fig. 7d). By changing the assumed direction of the rock flux, we are simply characterizing the change in the land surface with a different metric. We characterize it in terms of a mean horizontal motion, rather than as a mean vertical motion.

In two dimensions, the concept of horizontal motion of the Earth's surface with respect to the underlying rock is somewhat more complicated. The complex geometry (Fig. 7e) of any catchment surface implies that there will never be perfect horizontal motion of that surface. Any given catchment has facets that dip in all directions (Fig. 7e), so pure, uniform, steady horizontal motion of a catchment is impossible. However, much as the catchment-averaged erosion rate is an average of a spatially variable quantity, the horizontal velocity can also be regarded as an average, where individual facets of the surface

lower and retreat at different rates, but the net result can be characterized by an average horizontal velocity. For example, Fig. 7e shows a catchment from the Western Ghats draining the great escarpment. Although the drainage dips dominantly to the SW, there are both channel reaches and hillslopes that dip in all directions, including north or east. If erosion rates in this catchment are higher in the steep escarpment regions in the NE, individual slopes might retreat in any direction or not at all, but the catchment as a whole will expand to the NE, thereby "retreating" in this direction. The "average horizontal velocity"

describes the regional motion of the surface by averaging all the individual slope changes. By parameterizing the problem in terms of a horizontal velocity of the surface with respect to its underlying rock, rather than a vertical erosion rate, we do not



change any assumptions regarding geomorphic processes, or cosmogenic nuclide production and transport, but we do characterize the change in the landscape with a more representative metric. One important remaining assumption is that all points within the catchment are eroding fast enough that radioactive decay does not contribute significantly to secular

equilibrium.

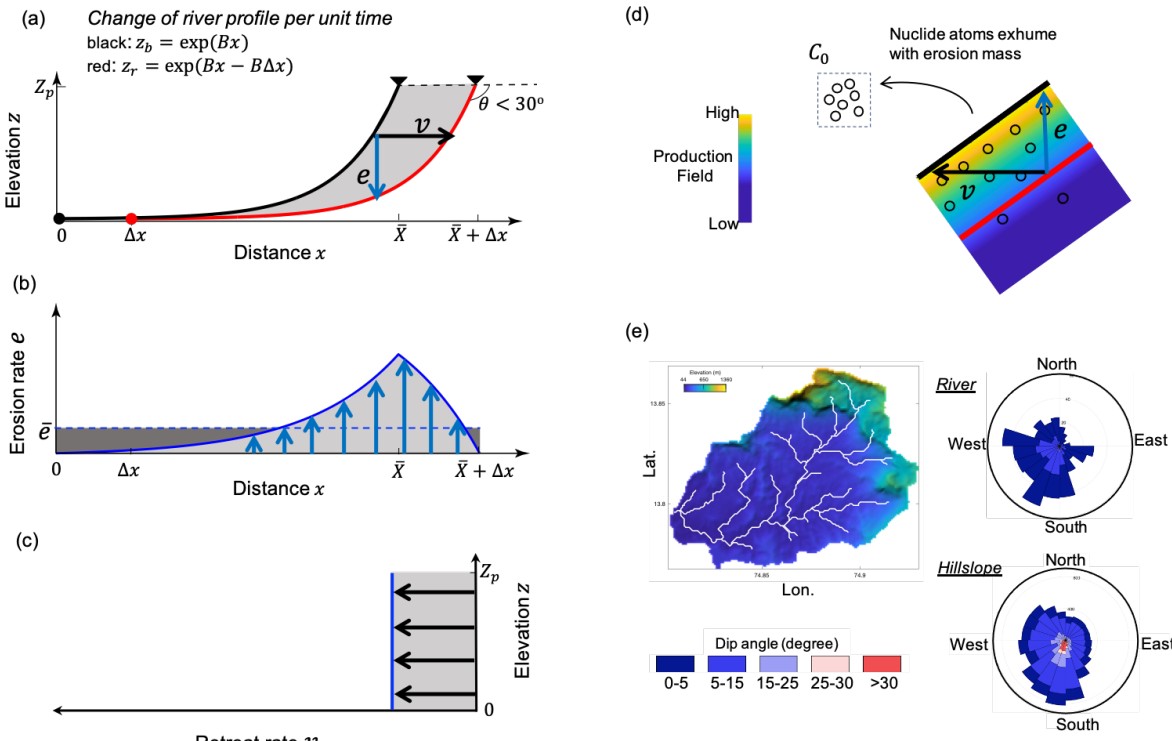

**Figure 7 Relationship between mass flux through a surface and the velocity for a retreating escarpment. (a) Land surface, assumed to be exponential in form, retreating into a pre-existing highland at constant velocity, *v*. The surface lowering at any point is given**

**as *e*. (b) Surface lowering rate as a function of distance. (c) Horizontal velocity as a function of elevation. Velocity is constant in both elevation and distance. Area under the curves (light and dark grey) in (a), (b) and (c) represents the mass flux and is equal in all cases. (d) CRN production zone below a dipping surface. Rock motion in either vertical or horizontal direction brings particles to the surface with an integrated production. For a constant flux, *v* and *e* scale so that the time of passage through the production zone is independent of direction, so surface concentration is independent of direction. (e) Elevation of a typical catchment in the**

**Western Ghats with rose diagrams showing frequency and magnitude of slope with direction for channels (upper) and hillslopes (lower).**

### 2.3.3 Mass flux in the horizontal direction

In this section, we derive expressions for catchment-wide cosmogenic nuclide production and concentration for arbitrary

direction of motion between rock and surface. We assume that there is no distortion of the surface or rock and that the





relative motion can be described as a single Euclidian vector without rotation. We present two methods for the calculation of mass flux and velocity from the measured detrital concentration and the catchment-wide nuclide production.

In the analysis above, we showed that the measured concentration of [10]Be in a sediment reflects the total integrated production in mass/time in the upstream catchment, divided by the rate of conversion of rock by erosion into sediment,

expressed as a rock mass flux. A measured [10]Be concentration gives a unique value for the rock mass flux, but not for the direction of velocity of the rock with respect to the surface. For the escarpment problem, the flux can be converted to a velocity by assuming horizontal rock motion. The calculation of velocity must take into account the complex shape of the catchment as well as uncertainty regarding the azimuthal direction of the rock velocity. We need to convert mass flux to a velocity in the appropriate direction normal to the escarpment, using an area projected normal to that direction. We present

two methods for doing the flux to velocity conversion here.

(1) Basin Projection Method

Consider a representative escarpment-draining basin (e.g. the basin in Fig. 6), if the erosion is completely efficient along the escarpment face, the escarpment would form a planar surface retreating horizontally, leaving a flat featureless coastal plain.

However, erosion is not completely efficient and although the channel profiles (Fig. 3) show localized relief onto the escarpment face, lateral variations are significant and catchments show considerable variability in morphology and presumably, erosion rate. The assumption in the calculation of an average is that inefficiencies in escarpment erosion are balanced by the continued erosion of remnant topography in the lowlands below the escarpment. During a unit time period, the mass of rock that is removed from the basin surface can be calculated from the retreat rate v and the area of the

catchment projected onto a vertical plane. The retreat vector, v, is normal to the vertical plane. Here we use $A_h$ to denote the projected area, and the conventional horizontal area is given as $A_v$ (Fig. 6). For a given flux, $V_{rock}$, we have the relationship:

$$eA_V = vA_h = V_{rock}, \tag{14}$$

Equation (14) expresses the relationship between velocity direction and projected area. Projection of a DEM-based

catchment surface onto a plane which is orthogonal to the projection direction gives a cluster of scattered points, including some that are identical in projected position. The projected area is given by the enclosed area of the outline that encloses these points on the projection plane.

(2) Local Scalar Product Method

As an alternative to full surface projection, we can also calculate the local surface projection. The dot product of the rock velocity and the catchment surface S in Eq. (9) can be calculated by the sum of the scalar product of all elemental surfaces with the mass flux vector:

$$F_s \cdot S \approx \sum_k \vec{F_s} \cdot \vec{n_k} A_k = |F_s| \sum_k \frac{\vec{F_s}}{|F_s|} \cdot \vec{n_k} A_k, \tag{15}$$





Where $\vec{n_k}$ and $A_k$ are the unit normal and the surface area of an elemental surface $k$ in the catchment surface S. Eq. (15)
states that the volume of eroded rock is the sum of the scalar product of elemental surfaces with the mass flux vector, and

equivalently, a sum of local mass flux from elemental surfaces (Fig. 8). The effective area is calculated from the sum of the

scalar product of the unit vector in the direction of $\vec{F_s}$ and elemental surfaces (Eq. (15)).

With a complex catchment surface, and a horizontal velocity, there are many local surfaces which dip towards the direction
of rock motion, which implies a local scalar product and a local mass flux $\vec{F_s} \cdot (\vec{n_k}A_k)$ that are negative (blue values in Fig.

8). These local negative values do not imply negative mass flux, but rather represent the multiple times that a vector in one

direction can cross the surface, $S$. Any local topographic high or transverse valley implies multiple intersections of a vector

with the surface, so any negative dot-product is compensated for by the multiple positive values.


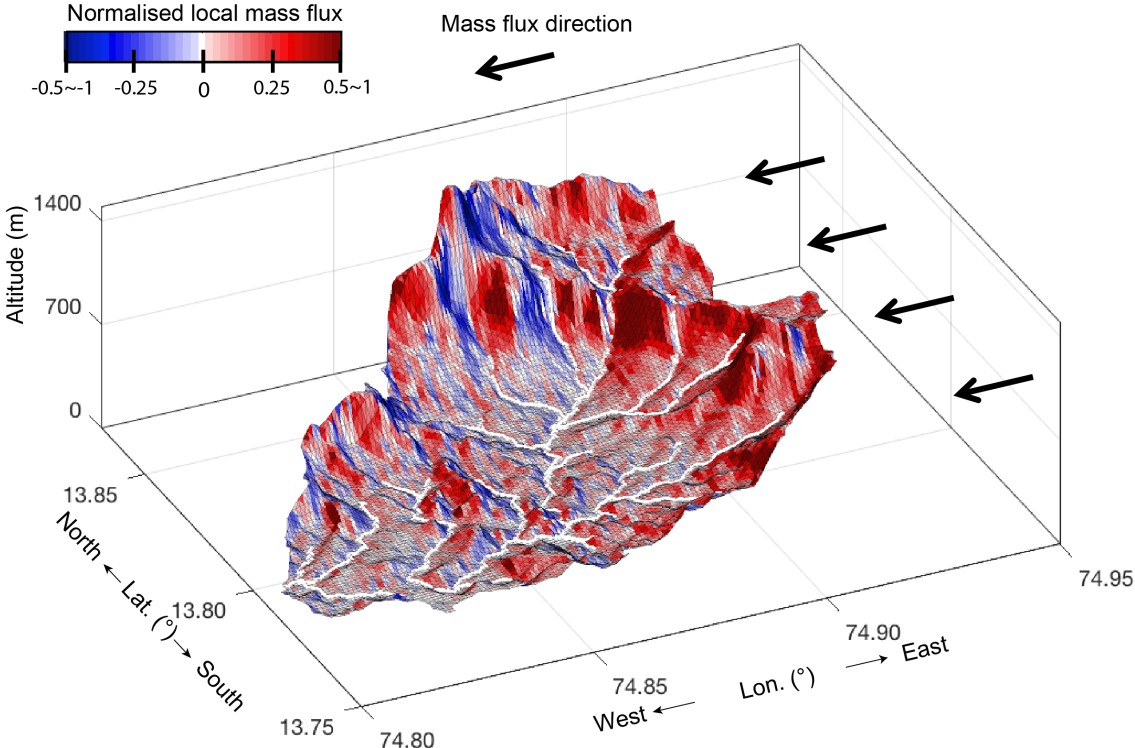

**Figure 8 Local mass flux of elemental surfaces of an escarpment-draining basin in Western Ghats. Basin location is indicated as Basin D in Fig. 2 and is also shown in Fig. 6. The basin surface is discretized into elemental surfaces and colour-coded with the relative magnitude of the local mass flux for a horizontal rock velocity in the direction indicated. Flux is normalized to the**
**catchment-averaged mass flux. Thick white lines are channels extracted from the DEM for a minimum drainage area of 1 km².**

(3) Azimuth of horizontal mass flux for escarpment retreat



In both the Basin Projection and Local Scalar Product methods, it is necessary to select a direction for the velocity or mass
flux vector. For the escarpment retreat problem, we can assume that this is purely horizontal, but with an unknown azimuth.
It is practical to sweep through a range of horizontal directions to determine the range of velocities associated with a range of
directions. We pick channels that drain through the escarpment but do not appear to be recent captures and take the
orientations of these channel segments as an estimate of the potential range of escarpment retreat directions. We then sweep
through all possible azimuths within this range. This is visualized by plotting the resulting vector magnitudes as a function of
directional azimuth.

An example of this calculation using each of these methods is shown in Fig. 9. The example basin is from the Western Ghats
with $^{10}$Be data published by Mandal et al. (2015). In each case, we test a range of azimuths from N20W to N85E. Inferred
horizontal velocities range from 180 m/Ma to 380 m/Ma, depending on azimuth.


The Local Scalar Product method has the characteristic that the inferred velocity increases rapidly as the direction rotates
towards parallel with the dominant catchment flow direction. This is an artefact of the ability of a scalar product to take
negative values. A typical catchment has a bowl-shaped geometry, open at the outlet, but as the basin is rotated, the
horizontal view eventually sees only the side of the bowl, and with further rotation, would provide a view of only the back of
the basin. Through this rotation, the effective area thus goes to zero or even negative, so that the flux goes to infinity for
these orientations.

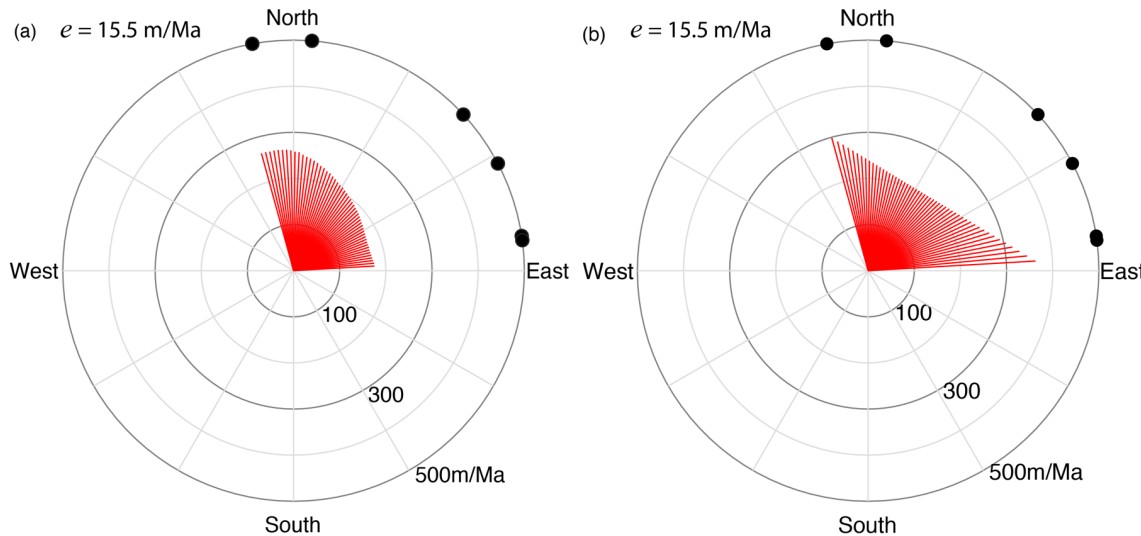

**Figure 9 Escarpment retreat rate as a function of azimuth of a horizontal mass flux vector using (a) the Basin Projection Method;**
**(b) the Local Scalar Product Method. Basin is from the Western Ghats with location indicated as Basin D in Fig. 2. Measured $^{10}$Be**
**concentration is 172426 (±8125) atoms/g and corresponds to a mean vertical erosion rate of 15.5 m/Ma. The azimuth of the**




**tangential red lines denotes the assumed escarpment retreat direction, and the length of the radial vector is the consequent retreat rate for that direction and the measured $^{10}$Be concentration. Black dots are orientations of characteristic tributaries of the escarpment-draining channels of this basin.**

**3. Applications and Verification**

As a demonstration as to how CRN data can be used to constrain the geomorphic evolution of a great escarpment, we use the cosmogenic $^{10}$Be data reported by Mandal et al. (2015) combined with our geomorphic analysis and proposed method of analysis in terms of horizontal flux. We have reprocessed the data of Mandal et al. (2015) for internal consistency, following the method and scaling relationships used in the method of Lupker et al. (2012) where the cosmogenic production rate is

calculated in a pixel-wise manner in comparison with the CRONUS method by Balco et al. (2008). Conventional erosion rates are also recalculated from the published DCN concentrations following the method of Lupker et al. (2012).

Mandal et al. (2015) report an average erosion rate of all escarpment-draining basins in southern Western Ghats to be 48.6 ± 20.9 m/Ma, which is low, but can be contrasted to the even lower erosion rates on the plateau to the east (~10 m/Ma). The

reported differential erosion rates, as well as the differential steepness across the continental drainage divide should drive the water divide migration towards the plateau (Mandal et al., 2015).

We evaluated all catchments draining the escarpment for which there are reported detrital $^{10}$Be data (Fig. 2). Each catchment was evaluated using each of the two methods described above; the radial plots summarizing the results are given with results

summarized in Tables (1) and (2). Fig. 10 shows an additional three examples of basins in the analysis. In all cases, catchments with erosion rates on the order of 10s of meters per million years transform into horizontal retreat rates of several hundred meters or a few kilometers per million years.

The two methods yield broadly consistent results (Table 2). However, the Local Scalar Product method is much more

sensitive to azimuth. It always shows a distinct minimum value of velocity, but velocity increases rapidly for other azimuths of velocity. For oddly shaped basins or for flux azimuths oblique to the dominant river direction, there is a strong influence of the negative values from the sides of a basin, and inferred velocities become large and deviate from those obtained from the Basin Projection method.

The basin geometry can play a large role in the inferred retreat rate. Fig. 10, basin (a1) shows an almost symmetrical basin, with a clear flow direction. The retreat rate is between 500 and 600 m/Ma and only weakly dependent on azimuth for the Basin Projection Method. The Local Scalar Product Method gives a similar result for the most likely retreat direction of N80E, but deviates quickly for other azimuths. In particular, a more northerly azimuth gives very high retreat rates, because high topography on the south margin of the basin provides many negative pixels, reducing the projected area and thus





increasing the flux. Fig. 10, basin (c1) is strongly asymmetric, with high topography (above ~800 m) in the NE. Here, the escarpment is clearly normal to the flow direction of the escarpment rivers (~ N30E). The Basin Projection Method gives retreat rates between 500 and 600 m/Ma, with a small variance over the likely range of azimuth. Similar to basin (a1), the higher topography on the plateau (above ~2000 m) provides many negative pixels, leading to high estimated velocities at northeastward azimuths. Fig. 10, basin (b1) is extremely asymmetric. Retreat rates from the Basin Projection Method are

however only weakly dependent on azimuth, suggesting a mean retreat rate of ~1100 m/Ma. The plateau is flat, although the shape is asymmetric, thus its projected area is small for any azimuth using the Basin Projection Method. The Local Scalar Product Method gives much larger retreat rates. Retreat direction of the escarpment is difficult to estimate from the flow directions as there are two major escarpment valleys oriented obliquely to each other. Surfaces of the four valley flanks intermittently give negative fluxes through rotation of various directions, leading to larger variability, but generally larger

retreat rates.

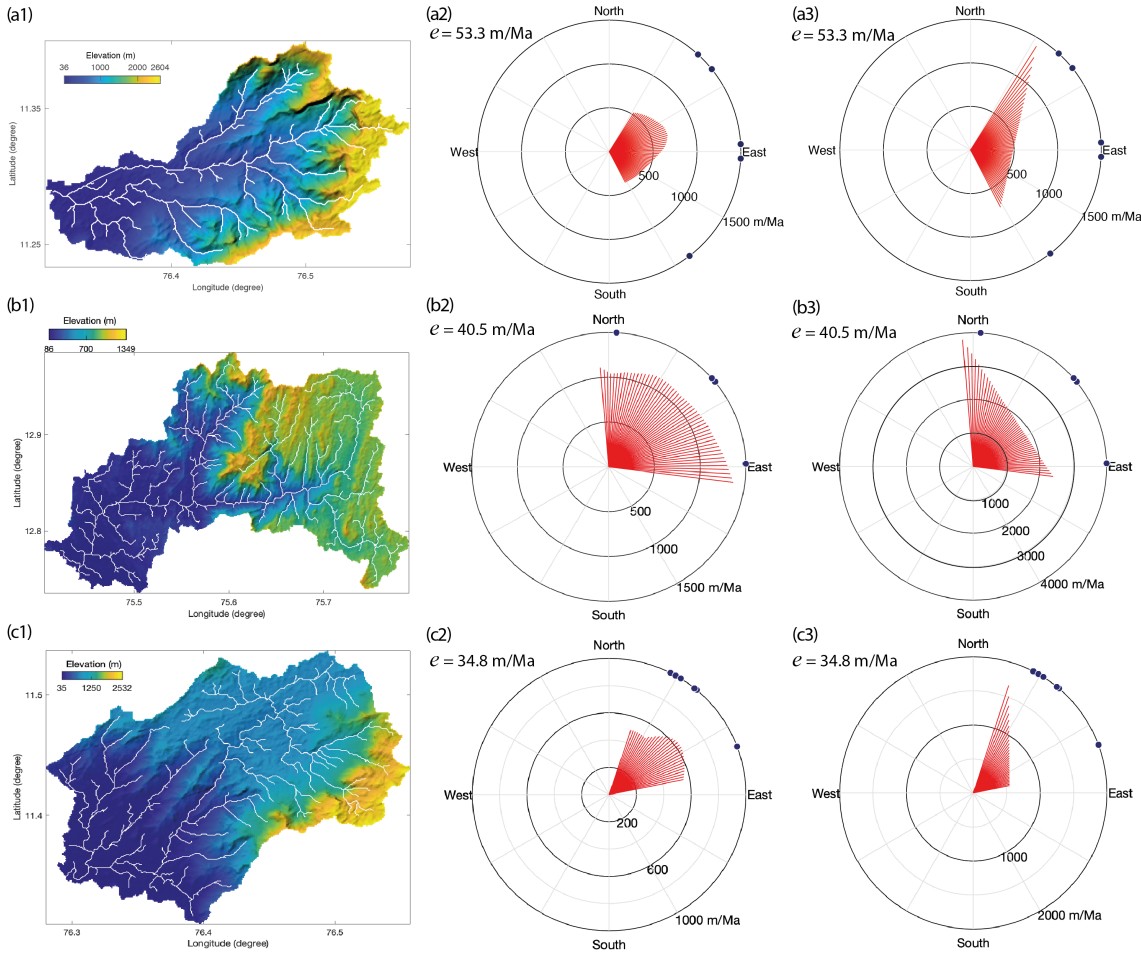





**Figure 10 Example basins from the Western Ghats showing the retreat rate as a function of azimuth of a horizontal mass flux vector using (a2, b2, c2) the Basin Projection Method; (a3, b3, c3) the Local Scalar Product Method. The azimuth of the red lines**
**denotes the assumed escarpment retreat direction, and the length of the radial vector is the consequent retreat rate for that direction and the measured [10]Be concentration. Black dots are orientations of characteristic tributaries of the escarpment-draining channels of this basin.**

In order to compare basins within the Western Ghats, we selected a common retreat direction and calculated retreat rates for
all basins in this direction. We selected a direction that is normal to the regional coast line, which is estimated to be trending at N158W. Escarpment retreat rates calculated from both methods vary from 171 m/Ma to 2427 m/Ma and are shown in map view in Fig. 11.

The average value of the retreat rates is close to the expected retreat rate expected from steady retreat of the escarpment from
the coast to its present location since the time of rifting of the margin (Fig. 12). The age of rifting of India from Madagascar is constrained to be between 120 Ma and 100 Ma (Thompson et al., 2019). The important event for the formation of the escarpment is the initial continental rifting that would have formed the new water divides at the crest of escarpments on each margin, so an earlier age is more likely. This event might have even predated other indications of rifting. The majority of retreat rates from [10]Be are lower than the post-rift average, although there are examples of higher rates. The lower rates
suggest that the escarpment retreat rate may have been higher in the early post-rift period. Alternatively, the retreat rate might have fluctuated in response to capture of rivers on the plateau surface, so that most observations are of a slower, background retreat rate that is periodically punctuated by more rapid retreat as a portion of the plateau area is captured. There is also significant spatial variation, and this variation co-varies with the shape of the escarpment creating a correlation between the retreat rate and the distance to the coastline (Fig. 12). For example, the large embayment at 13N latitude
corresponds to the highest modern retreat rates. This suggests that the modern rates have been sustained since the time of rifting and variations reflect long-lived characteristics of the escarpment or the retreat processes.





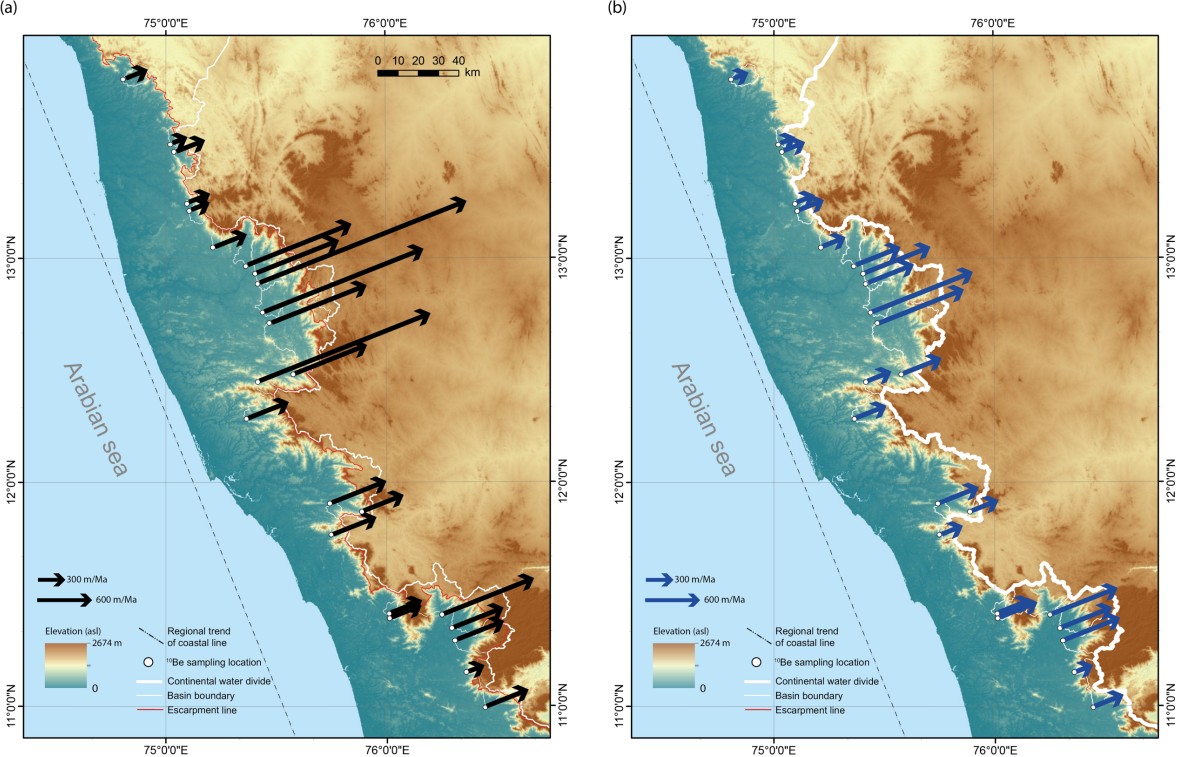

**Figure 11 Escarpment retreat rates in the normal direction of the reference coastal line on the topography base map of southern Western Ghats. Rates from (a) the Local Scalar Product method and (b) from the Basin Projection method. Arrows represent the escarpment retreat vector: the azimuth denotes the retreat direction while the length denotes the retreat rate in that direction. The black dashed line indicates the trend of the modern coastline and escarpment.**





**Figure 12 Current distance from the coastline against inferred retreat rate of Western Ghats escarpment basins from detrital [10]Be concentrations with 1 σ uncertainty. Retreat rates of the escarpment are calculated using the Basin Projection method with azimuth taken as N57E. See Table 2 for the data. Age of rifting is constrained by various events as indicated and is expected to be at older than these constraints. Basins smaller than 50 km² are indicated with open red circles.**

We also investigated the relationship between channel steepness, escarpment elevation and escarpment retreat rate (Fig. 13, 14). Steepness is correlated with relief or total height of the escarpment (Fig. 13). Such a relationship was predicted by Willett et al. (2018) for escarpments retreating at a constant velocity; steepness would need to increase with height in order to maintain the higher velocity associated with higher remnant topography. We also plot the predicted relationship between steepness and retreat rate from Eq. (2) for two values of slope exponent, *n* and erodibility, *K* (Fig. 14). A range of *K* would be necessary to explain the full scatter, but it is also possible that some of the scatter is associated with plateau river capture and transience in the river profiles.



Earth **Surface**
**Dynamics**
Discussions

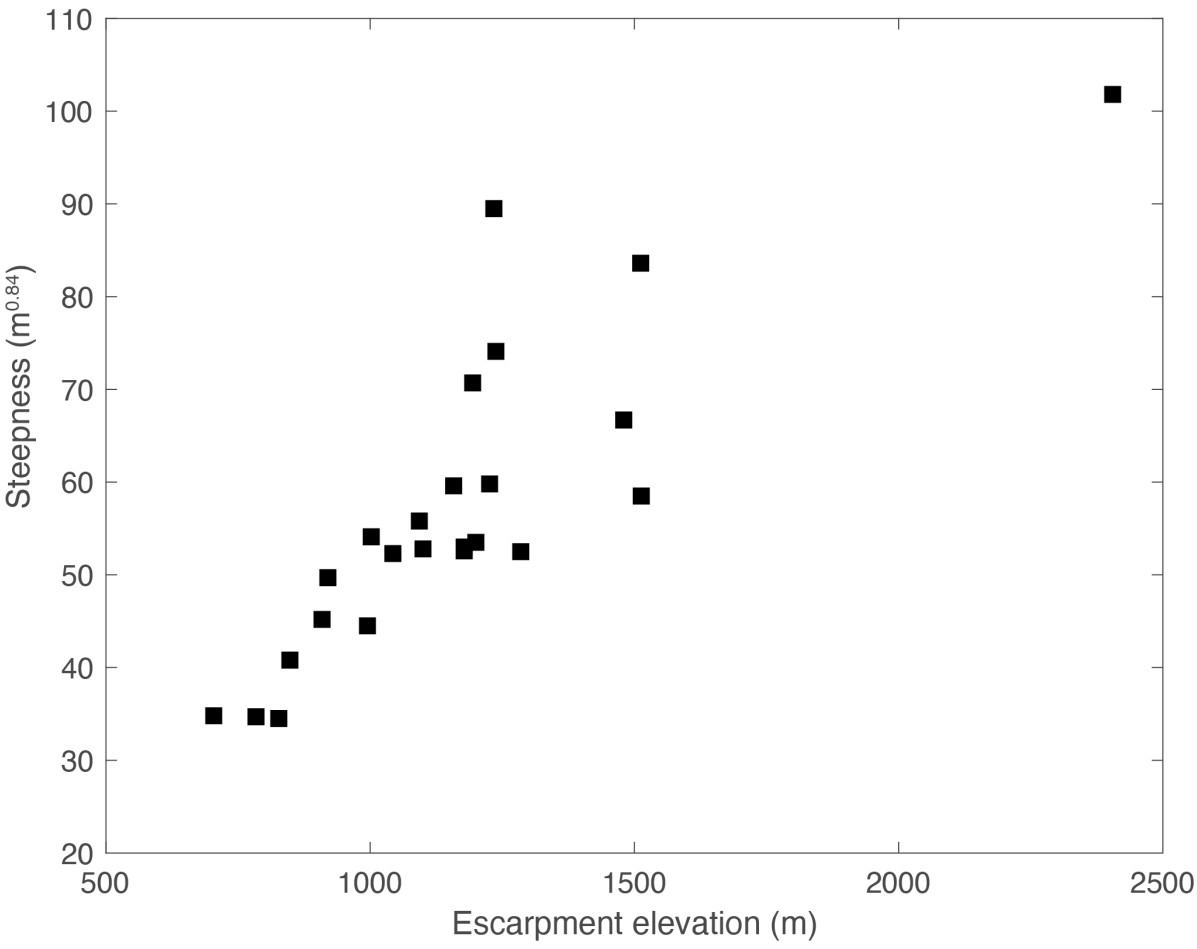

**Figure 13 Relationship between channel steepness and escarpment height in the Western Ghats. The channel steepness is calculated from regression of slope-area plots using a uniform concavity of 0.42. Correlation is consistent with the theory that morphology has evolved to erode pre-existing topography at a constant rate of retreat.**




Earth **Surface**
**Dynamics**
Discussions

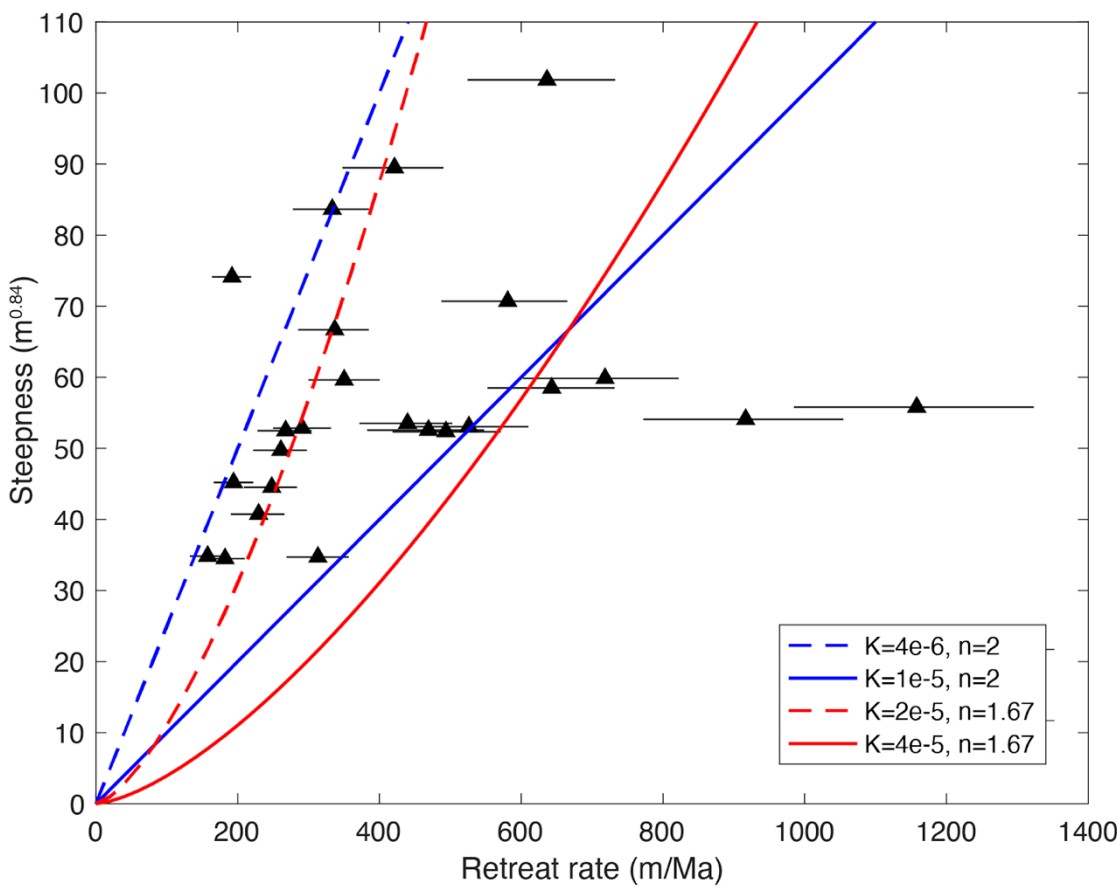

**Figure 14 Channel steepness against escarpment retreat rates for catchments in the Western Ghats. Retreat rates of the**
**escarpment are calculated using the Basin Projection method with an azimuth of N57E. Short solid lines indicate 1 σ uncertainty**
**of retreat rates. Lines are model prediction of steepness and retreat rates from Eq. (2) for a range of the slope exponent *n* and *K***
**where the dimensions of *K* are** $\frac{m^{1.84-0.84n}}{yr}$.

## Table 1 Cosmogenic [10]Be basins of Western Ghats studied in this research

| Published Basin (a) | Lat. (°) | Lon. (°) | Cosmogenic [10]Be concentration(atoms/g) (b) | | Sample ele. (m) | Esp. ele. (m) | Mean slope(°) (d) | Slope >30° (%) (e) | Published erosion rate (m/Ma) | | Recalculated erosion rate (m/Ma) (f) | | |
|---|---|---|---|---|---|---|---|---|---|---|---|---|---|
| | | | Concentration | Error(±) | | | | | Rate | Uncertainty (±) | Rate | +34% | -34% |
| SIN1440 | 13.7960 | 74.8034 | 172426 | 8125 | 44 | 827 | 12 | 6 | 19.67 | 1.63 | 15.53 | 2.35 | 2.31 |
| SIN1380 | 13.5069 | 75.0212 | 133454 | 9211 | 83 | 704 | 12 | 7 | 29.17 | 2.83 | 20.35 | 3.14 | 3.23 |
| SIN1435 | 13.2432 | 75.0966 | 102250 | 5223 | 72 | 909 | 15 | 8 | 39.10 | 3.22 | 29.50 | 4.23 | 4.27 |
| SIN1433 | 13.2100 | 75.1062 | 90233 | 5261 | 72 | 995 | 12 | 4 | 41.57 | 3.61 | 31.15 | 4.49 | 4.94 |
| SIN1432 | 13.0480 | 75.2141 | 70145 | 4074 | 104 | 1285 | 14 | 7 | 60.78 | 5.21 | 45.38 | 6.20 | 6.70 |





| SIN1369 | 12.9638 | 75.3649 | 77251 | 6585 | 77 | 1178 | 12 | 6 | 55.53 | 5.98 | 37.14 | 5.93 | 6.08 |
|---|---|---|---|---|---|---|---|---|---|---|---|---|---|
| SIN1368 | 12.9295 | 75.4067 | 70502 | 5028 | 78 | 1226 | 13 | 4 | 64.80 | 6.20 | 43.42 | 6.28 | 6.99 |
| SIN1367 | 12.8833 | 75.4181 | 73573 | 8296.5 | 107 | 1178 | 15 | 9 | 64.19 | 7.29 | 41.21 | 6.89 | 7.61 |
| SIN1362 | 12.7566 | 75.4401 | 80104 | 5184 | 86 | 1093 | 11 | 4 | 59.14 | 5.38 | 40.46 | 5.77 | 6.05 |
| SIN1361 | 12.7098 | 75.4695 | 90315 | 6672 | 91 | 1002 | 12 | 4 | 48.33 | 4.76 | 32.81 | 4.93 | 5.16 |
| SIN1430 | 12.4804 | 75.5794 | 69430 | 5001 | 153 | 1200 | 16 | 6 | 66.68 | 6.44 | 49.82 | 7.15 | 7.70 |
| SIN1427 | 12.4466 | 75.4186 | 94374 | 5389 | 167 | 1100 | 18 | 4 | 47.87 | 4.13 | 36.95 | 5.16 | 5.21 |
| SIN1429 | 12.2815 | 75.3675 | 97093 | 4932 | 42 | 1158 | 18 | 8 | 41.08 | 3.36 | 31.10 | 4.47 | 4.48 |
| SIN1419 | 11.9043 | 75.7418 | 106563 | 6034 | 76 | 1043 | 12 | 3 | 32.47 | 2.80 | 24.43 | 3.83 | 3.72 |
| SIN1421 | 11.8677 | 75.8892 | 107228 | 5571 | 196 | 784 | 17 | 13 | 41.72 | 3.47 | 32.50 | 4.52 | 4.60 |
| SIN1416 | 11.7651 | 75.7500 | 109089 | 6025 | 69 | 920 | 20 | 17 | 37.04 | 3.16 | 28.48 | 4.01 | 4.26 |
| SIN1350 | 11.4120 | 76.0106 | 74345 | 6593 | 37 | 1512 | 15 | 13 | 64.24 | 7.09 | 42.31 | 6.55 | 7.05 |
| SIN1351 | 11.3913 | 76.0130 | 71843 | 8383 | 31 | 1234 | 21 | 24 | 76.26 | 10.36 | 53.75 | 8.81 | 9.37 |
| SIN1331 | 11.4070 | 76.2507 | 82246 | 7018 | 45 | 1194 | 15 | 9 | 64.08 | 6.92 | 44.77 | 6.47 | 7.22 |
| SIN1332 | 11.3473 | 76.2947 | 110089 | 7370 | 35 | 1513 | 13 | 6 | 48.57 | 4.57 | 34.77 | 4.78 | 4.89 |
| SIN1330 | 11.2890 | 76.3074 | 86695 | 9867 | 62 | 2405 | 20 | 27 | 77.32 | 10.35 | 53.32 | 8.05 | 9.40 |
| SIN1407 | 11.1503 | 76.3591 | 84821 | 4828 | 85 | 1238 | 21 | 20 | 48.74 | 4.17 | 36.79 | 5.18 | 5.37 |
| SIN1348 | 10.9917 | 76.4440 | 84696 | 7081 | 65 | 1480 | 17 | 16 | 69.46 | 7.42 | 49.79 | 7.05 | 7.63 |
| SIN1379 | 13.4731 | 75.0358 | 198955 | 18234 | 90 | 848 | 13 | 7 | 20.70 | 2.45 | 15.19 | 2.41 | 2.59 |

(a) Basin name is from published DCN [10]Be sample name.
(b) DCN [10]Be concentration data are from Mandal et al. (2015).
(d) Slope is calculated from TopoToolBox function gradient8.m of surface pixels.
(e) The percentage of surface pixels steeper than 30°, the slope is calculated from TopoToolBox function gradient8.m of surface pixels.
(f) For internal consistency, conventional erosion rates are recalculated from reported [10]Be concentrations following method
of Lupker et al. (2012). The uncertainty of ±34% are propagated from the measured error of [10]Be concentration.

**Table 2 [10]Be-inferred escarpment retreat rates of Western Ghats, India**

| Published Basin (a) | Steepness (m^0.84) | Distance to coastline (km)(g) | Retreat rate of minimum dot retreat direction (m/Ma) | | | | | | Retreat rate of regional escarpment retreat direction (m/Ma) | | | | | |
|---|---|---|---|---|---|---|---|---|---|---|---|---|---|---|
| | | | Direction (h) | Local Scalar Product Method | | | Basin Projection Method | | Direction (h) | Local Scalar Product Method | | | Basin Projection Method | | |
| | | | | Rate | +34% | -34% | Rate | +34% | -34% | | Rate | +34% | -34% | Rate | +34% | -34% |
| SIN1440 | 34.5 | 32.2 | 30.7 | 210 | 32 | 31 | 213 | 32 | 32 | 57 | 233 | 35 | 35 | 182 | 28 | 27 |
| SIN1380 | 34.8 | 44.2 | 53.0 | 174 | 27 | 28 | 153 | 24 | 24 | 57 | 175 | 27 | 28 | 158 | 24 | 25 |
| SIN1435 | 45.2 | 46.2 | 50.6 | 243 | 35 | 35 | 200 | 29 | 29 | 57 | 245 | 35 | 35 | 194 | 28 | 28 |
| SIN1433 | 44.5 | 48.6 | 68.1 | 218 | 31 | 35 | 293 | 42 | 46 | 57 | 223 | 32 | 35 | 248 | 36 | 39 |
| SIN1432 | 52.5 | 55.6 | 19.0 | 290 | 40 | 43 | 350 | 48 | 52 | 57 | 334 | 46 | 49 | 268 | 37 | 40 |
| SIN1369 | 53.0 | 69.4 | 14.2 | 739 | 118 | 121 | 474 | 76 | 78 | 57 | 1007 | 161 | 165 | 526 | 84 | 86 |





| SIN1368 | 59.8 | 79.7 | 54.4 | 951 | 138 | 153 | 707 | 102 | 114 | 57 | 952 | 138 | 153 | 718 | 104 | 116 |
| SIN1367 | 52.6 | 83.6 | 359.1 | 857 | 143 | 158 | 461 | 77 | 85 | 57 | 1655 | 277 | 306 | 469 | 78 | 87 |
| SIN1362 | 55.8 | 86.3 | 56.5 | 1847 | 263 | 276 | 1157 | 165 | 173 | 57 | 1847 | 263 | 276 | 1158 | 165 | 173 |
| SIN1361 | 54.1 | 79.3 | 104.1 | 914 | 137 | 144 | 1054 | 158 | 166 | 57 | 1340 | 201 | 211 | 917 | 138 | 144 |
| SIN1430 | 53.5 | 65.9 | 92.5 | 788 | 113 | 122 | 496 | 71 | 77 | 57 | 964 | 138 | 149 | 440 | 63 | 68 |
| SIN1427 | 52.8 | 48.0 | 143.3 | 535 | 75 | 75 | 454 | 64 | 64 | 57 | 6964 | 972 | 982 | 291 | 41 | 41 |
| SIN1429 | 59.6 | 47.6 | 59.4 | 478 | 69 | 69 | 352 | 51 | 51 | 57 | 478 | 69 | 69 | 350 | 50 | 50 |
| SIN1419 | 52.3 | 34.2 | 71.9 | 576 | 90 | 88 | 440 | 69 | 67 | 57 | 1083 | 170 | 165 | 494 | 77 | 75 |
| SIN1421 | 34.7 | 47.5 | 131.5 | 921 | 128 | 130 | 329 | 46 | 47 | 57 | 503 | 70 | 71 | 313 | 44 | 44 |
| SIN1416 | 49.7 | 32.8 | 76.3 | 527 | 74 | 79 | 258 | 36 | 39 | 57 | 557 | 78 | 83 | 261 | 37 | 39 |
| SIN1350 | 83.6 | 43.0 | 58.0 | 375 | 58 | 62 | 338 | 52 | 56 | 57 | 375 | 58 | 62 | 333 | 52 | 56 |
| SIN1351 | 89.5 | 45.2 | 86.3 | 358 | 59 | 62 | 502 | 82 | 88 | 57 | 404 | 66 | 70 | 421 | 69 | 73 |
| SIN1331 | 70.7 | 57.7 | 271.5 | 989 | 143 | 160 | 756 | 109 | 122 | 57 | -1208 | 195 | 175 | 581 | 84 | 94 |
| SIN1332 | 58.5 | 75.0 | 72.4 | 579 | 80 | 82 | 589 | 81 | 83 | 57 | 656 | 90 | 92 | 643 | 88 | 90 |
| SIN1330 | 101.8 | 83.1 | 102.7 | 478 | 72 | 84 | 499 | 75 | 88 | 57 | 683 | 103 | 120 | 636 | 96 | 112 |
| SIN1407 | 74.1 | 67.8 | 62.6 | 193 | 27 | 28 | 198 | 28 | 29 | 57 | 195 | 27 | 28 | 192 | 27 | 28 |
| SIN1348 | 66.7 | 69.7 | 29.7 | 402 | 57 | 62 | 364 | 52 | 56 | 57 | 451 | 64 | 69 | 337 | 48 | 52 |
| SIN1379 | 40.8 | 47.5 | 102.4 | 292 | 46 | 50 | 283 | 45 | 48 | 57 | 417 | 66 | 71 | 230 | 36 | 39 |

(g) Distance is calculated using a referential retreating direction that is perpendicular to the regional escarpment.

(h) The angle is defined as clockwise from the geographic North (0º).

## 4. Discussion

### 4.1 Correction for flexural isostatic rebound on ¹⁰Be-inferred retreat rate

One effect not accounted for in our horizontal flux calculation is the vertical uplift and flux that results from flexural compensation of the mass eroded from the escarpment face. Retreat of the escarpment generates a flexural isostatic uplift
centered at the escarpment front, but spread over a distance that encompasses the nearby plateau and lowland coastal plain. Approximately half of the flexural uplift occurs on the plateau side of the escarpment and is thus accounted for in the horizontal retreat calculation (Fig. 15a), but uplift of the coastal plain is not and if this uplift is eroded, it represents a vertical component to the erosional flux that we have not accounted for. Not accounting for this eroded mass implies that we have overestimated escarpment retreat rates, so we assess how large this effect is here.


The flexural uplift rate due to escarpment erosion can be quantified with the isostatic deflection of a simple line load centered on the escarpment. The magnitude of the line load is expressed as:

$$N_0 = v\Delta t H g \rho_{crust},\qquad(16)$$

where $N_0$ is the load, and is a function of the escarpment height, $H$, and the retreat rate, $v$ (Fig. 15a). As we are interested in
the velocity, not the cumulative uplift, this is done for a small time, $\Delta t$. The density of eroded rock is given by $\rho_{crust}$. The deflection of this line load is given by (Turcotte and Schubert, 2002):



$$w(x) = w_{max}\exp\left(-\tfrac{x}{\alpha}\right)\left(\cos\left(\tfrac{x}{\alpha}\right) + \sin\left(\tfrac{x}{\alpha}\right)\right), \tag{17}$$

where $w_{max}$ the is the maximum deflection uplift, $\alpha$ is the characteristic wavelength of flexural deflection, given as:

$$W_{max} = \frac{N_0}{2} * \frac{1}{\Delta\rho g\alpha} = \frac{v\Delta tHg\rho_{crust}}{2} * \frac{1}{\Delta\rho g\alpha} = \frac{v\Delta tH\rho_{crust}}{2\Delta\rho\alpha}, \tag{18}$$


$$\alpha = \left\{\frac{4D}{\Delta\rho g}\right\}^{1/4}, \tag{19}$$

where $\Delta\rho$ is the density contrast between mantle and air; and $D$ is the flexural rigidity (Turcotte and Schubert, 2002):

$$D = \frac{ET_e^3}{12(1-\zeta^2)}, \tag{20}$$

where $\zeta$ is Possion's ratio, $E$ is Young's modulus and $T_e$ is the effective elastic thickness that characterizes the lithosphere
rigidity.

The effect on our calculations depends on the uplift between the escarpment and the point at which a detrital CRN sample is taken. In Figure 15, we show this at the coast, but in practice this will be closer to the escarpment. The integrated uplift from the escarpment to the sample point at $X_c$ is:

$$\int_0^{X_c} w\,dx = w_{max}\int_0^{X_c}\exp\left(-\tfrac{x}{\alpha}\right) * \left[\cos\left(\tfrac{x}{\alpha}\right) + \sin\left(\tfrac{x}{\alpha}\right)\right]dx, \tag{21}$$

Solving this gives an area of eroded material:

$$A_{uplift} = -\alpha w_{max}\left(\exp\left(-\tfrac{X_c}{\alpha}\right)\cos\left(\tfrac{X_c}{\alpha}\right) - 1\right), \tag{22}$$

The escarpment retreat implies an area of erosion of:

$$A_{retreat} = v\Delta tH, \tag{23}$$

The missing area due to flexural uplift can be quantified by the ratio of uplifted area to the retreat area:

$$R_A = \frac{A_{uplift}}{A_{retreat}} = \frac{\rho_{crust}}{2\rho_{mantle}} * \left(\exp\left(-\tfrac{X_c}{\alpha}\right)\cos\left(\tfrac{X_c}{\alpha}\right) - 1\right), \tag{24}$$

The density contrast between mantle and the upper crust can be constrained to a range of (0.78~0.92) with an average ratio of 0.85 at global scale (Tenzer et al., 2012).

Fig. 15b shows the area ratio $R_A$ for a range of $X_c$ and the effective elastic thickness at a density contrast of 0.85 between upper crust and mantle. The eroded area ratio $R_A$ decreases as the effective elastic thickness of the lithosphere increases. At a mature passive margin, the lithosphere effective elastic thickness, $T_e$ is typically a few 10s of km (Audet and Bürgmann, 2011). For the Western Ghats data, most samples are within 20 km of the centre of the escarpment (Fig. 3). At a $T_e$ = 20 km, the mass area ratio $R_A$ is below 15% for $X_c$ = 20 km (Fig. 15b). This error is partially offset by the rock uplift that establishes
the coastal plain slope, which for this calculation, we assume is flat.





**Figure 15 (a) Schematic model of isostatic flexural uplift due to escarpment erosion and retreat. Flexural uplift is calculated assuming removal of mass $v\Delta t$ treated as a line load at $x = 0$; (b) Area ratio showing vertical erosion of coastal plain not accounted for in our horizontal retreat model. Ratio represents an overestimate of our retreat rates and is up to a few tens of percent at small $T_e$ if sample is taken distant from escarpment.**

## 4.2 Methods for converting flux to velocity

The calculation of a horizontal escarpment retreat rate is based on recognition that the concentration of $^{10}$Be in a detrital sediment is proportional to a flux of rock from the Earth, regardless of direction or spatial variation of local flux. Cosmogenic $^{10}$Be concentration provides a measure of the flux of rock from the earth, but this need not be an erosion rate in



the sense of a vertical velocity. Measurement of the surface area, projected onto a horizontal plane, can be used to convert the flux into a vertical velocity, which is defined to be the erosion rate, which is what is done in a conventional analysis. The calculation of a horizontal retreat rate is an identical process. A direction must be selected and the surface of the catchment

projected onto a plane perpendicular to this direction. The horizontal retreat rate depends on the effective area of a basin in the migration direction. We present two methods to calculate the effective area. The Basin Projection Method calculates the projected area of the overall basin surface onto a vertical plane and the Local Scalar Product Method calculates the effective area by summation of the dot-product of local elemental surface normal with the velocity. If an escarpment were a perfect, uniform plane, these methods would give identical predictions, but for real landscapes, deviations of the land surface from a

single plane results in differences in the velocity calculation, based on how the normal to the surface is calculated.

In practice, there are some disadvantages to the Local Scalar method. Primary is the strong effect of blocking of flux by neighboring basins. If the flux direction is such that the sides of the basin overlap with neighboring basins, a negative flux results from these sides, cancelling the corresponding flux from the escarpment surface. This is the "edge of the bowl"

problem discussed above, where if one viewed the basin in the direction of the flux vector, one could see only the outside of the basin. This is not necessarily an error, depending on the geometry of the basins and the erosive role of the neighboring basin, this may or may not be correct, but the basin projection method is less sensitive to this effect, and so gives a more robust, if not more accurate, result.

### 4.2.1 Remnant topography in an escarpment-draining basin

Within a given escarpment-draining basin, the dependence of retreat rate on retreat direction (e.g. Fig. 9) comes directly from the geometry of the basin surface. Isolated buttes, inselbergs or other topography internal to the drainage basin such as escarpment-normal interfluvial ridges, define a type of remnant topography due to inefficient retreat of the escarpment. Buttes are generally regarded to be part of the ancient escarpment, but now are erosion residuals (Gunnell and Harbor, 2010). Their existence attests to some inefficiency in the past escarpment erosion, such that a portion of the high topography is not

removed as the escarpment passes. If the erosional efficiency of the escarpment is variable along strike, we would expect the low efficiency segments of the escarpment to lag, and potentially become isolated from the escarpment forming a butte. An important question for our analysis is how does this impact the [10]Be concentrations and inferred retreat rates?

By calculating the mean retreat rate, we are implicitly assuming that anomalously slow retreating escarpment segments,

including incipient buttes, are balanced by segments elsewhere in the same catchment that have higher retreat rates. As with vertical erosion rate calculations, spatial variation will not impact the mean unless it has an extreme variation that affects the assumptions regarding production or radioactive decay. This is also valid for the formation of remnant topography. For example, a butte forms because part of the escarpment was eroding at a retreat rate slower than the average. After a butte has formed as an isolated topographic feature, it continues to erode, thus contributing sediment (and [10]Be atoms) to the net total





of the basin. The principle of the average rate is that the extra sediment coming from the butte balances the "missing" sediment that does not arrive from a slow-eroding segment of the current escarpment. Provided the rate of remnant topography formation remains constant, the variance of retreat rate on the escarpment remains constant, and if the basin is large enough to average this process, we will obtain the correct average $^{10}$Be concentration and the correct retreat rate.

### 4.2.2 Retreat direction

Pure horizontal mass flux is an end-member flux direction (the other end-member is purely vertical) still has its azimuthal direction to be determined. Both methods for calculating a purely horizontal mass flux have an important azimuthal dependence. The Local Scalar Product method is more sensitive than the Basin Projection method to the assumed direction. Drainage basins have surfaces dipping in all directions but dominant directions are evident (e.g. Fig. 7e). Basins with a well-defined regional slope and form yield consistent rates for azimuths near the dominant slope direction. Inferred rates deviate

if the azimuth varies far from the dominant direction, particularly with the Local Scalar Product method. The Basin Projection method displays less azimuthal sensitivity.

A rift margin escarpment is sinuous, implying variations in its average long-term retreat rate and local direction. The Western Ghats is assumed to be retreating at N57E, but locally, there is likely to be considerable variation in retreat

direction.

### 4.2.3 Temporal variations in retreat rates

Retreat rates from our analysis are on the order of 100 m/Ma to 1 km/Ma. The retreat rates are horizontal, but the dominant physical process is still vertical incision of rivers (Fig. 7 a, b, c). As we use DCN $^{10}$Be concentrations for retreat rate calculations, these retreat rates are valid through the vertical erosion of one attenuation length of rock (Fig. 7d), providing an

integration time for the measurement. In the Western Ghats, the basin-averaged erosion rate of escarpment-draining basins is 10s-100s of m/Ma. An erosion rate of 100s of m/Ma implies an integration time of ~$10^3$ years, which means the escarpment retreat rates are integrating over a thousand-year timescale.

Using the current coastline and the rifting age as reference position and time, also gives an average retreat rate of

100s~1000s of m/Ma since continent break up. For this calculation, the modern coastline is assumed to represent the locus of the breakup structures, which may not always be true, although the western margin of India shows that the structural hinge between uplift and subsidence is close to the modern coast (Campanile et al., 2008; Chaubey et al., 2002). Average retreat rates are in the same range of magnitude as of the DCN $^{10}$Be-inferred retreat rates (Fig. 12), although they are systematically higher, suggesting that the modern escarpment is moving slower than its historical average. This difference becomes larger if

we correct our retreat rates for flexural compensation. However, the majority of points are within a  factor of 2 for an initiation of rifting at 120 Ma. Small escarpment basins have a higher probability of reflecting local effects, e.g. resistant



lithology, structural or metamorphic fabrics and show larger variance. Taking these small basins out of the analysis, the Western Ghats has a long-term retreat rate well represented by the DCN $^{10}$Be-inferred rate.

Although offshore sedimentation records are difficult to interpret given that basins are open to sediment export and recycling of sediment by subsequent erosion, records do not generally support steady rates of sediment supply with time since rifting. The offshore Konkan and Kerala Basin abutting the Western Ghats record two pulses of intensive sedimentation: a Paleocene phase and a Pliocene phase (Campanile et al., 2008).  If these sediment records are correct reflections of sediment supply from the eroding escarpment, it suggests that the correspondence between modern and historical retreat rates might be

fortuitous. However, the cause of these variations in erosion rate remains open. Escarpment relief might have changed over time if there has been significant continental uplift or tilting due to mantle dynamic flow, but there is no evidence for this aside from the variations in offshore sediment volume. The consistency in escarpment morphology and lack of along-strike variations in height, morphology or distance from the coast, suggests that escarpment retreat has been the dominant process since rifting and any dynamic uplift would need to affect the entire margin nearly uniformly. This, however, does remain

possible and would affect the temporal variability of escarpment retreat. The other possibility is climate change; changing precipitation rates through the Cenozoic would also affect the erosional efficiency and thus retreat rate of the escarpment. Given India's migration from the tropics to its current mid-latitude position and global climate changes over the Cenozoic (Kent and Muttoni, 2008), climate change certainly occurred and some impact on temporal variations in retreat rate is likely.

## 5. Conclusions

Large escarpments such as occur on many passive margins represent disequilibrium landscapes that have a long timescale of response and are characterized by erosion rates localized onto the escarpment, thereby driving retreat of the escarpment inland. Erosion rates are strongly variable in space, so an average erosion rate, as derived from detrital cosmogenic nuclides for catchments draining the escarpment, does not give an effective characterization of the rates of landscape change. We have addressed this by showing how $^{10}$Be concentrations can be used to measure the average rate of escarpment retreat.

Retreat rates obtained from $^{10}$Be data from the Western Ghats of India gave retreat rates of up to more than 2 km/Ma. These rates are broadly consistent with the distance of the escarpment from the coast, suggesting that they are representative of the long-term rates.

Study of the Western Ghats demonstrates that the morphology of the escarpment rivers is consistent with evolution of the

escarpment to a form driving escarpment retreat at a constant rate with a low-steepness coastal reach keeping up with sediment transport and isostatic uplift and a steep-escarpment reach driving landward retreat. Escarpment retreat leads to episodic capture of plateau rivers, and we found numerous examples of rivers with high flat reaches characteristic of capture.



These examples were distributed randomly along the escarpment, inconsistent with the alternative model of constant catchment geometry and transient uplift.


The general conclusion of this study is that great escarpments on passive margins are dynamic features with significant rates of retreat and high local rates of mass removal by erosion. Although rates are likely to be variable in time, escarpment retreat appears to be active from the time of rifting to the modern with current and average rates at hundreds to thousands of meters per million years, and these rates can be estimated by analysis of cosmogenic isotope concentrations.


*Code availability.* Related codes are available upon request to the correspondence author.

*Author contributions.* YW and SW conceptualized and designed the research. YW analyzed morphological features, developed the theory, developed the codes, conducted calculations, analyzed the data, wrote the manuscript and prepared the figures. SW contribute to theory development, data interpretation and provide input on the manuscript.

*Competing interests.* The authors declare that there is no conflict of interest.

*Acknowledgements.* We thank Maarten Lupker for valuable discussions.

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
