# Peer review of "Escarpment retreat rates derived from detrital cosmogenic nuclide concentrations"

_Earth Surface Dynamics, 2021_

## Author Response (AR1)

Response to Anonymous Referee #1

**Reviewer:**
Wang and Willett present two methods for converting basin-averaged cosmogenic nuclide derived vertical erosion rates to horizontal escarpment retreat rates, and they use these methods to infer retreat rates in catchments along the escarpment of the Western Ghats. They find that these millennial timescale retreat rates are in the same range as Myr timescale retreat rates inferred from the age of rifting and distance of the escarpment from the coast, suggesting a near constant rate of retreat since rifting. Congruent with this observation, they also find that the morphology of escarpment rivers and the scaling between channel steepness and retreat rates are consistent with landscape evolution via constant retreat of a steep escarpment and minor subsequent evolution of the low steepness coastal plain.

I found the manuscript to be well written and illustrated, with a clear and well motivated introduction/problem statement and well conceived analyses to present the methodology and results of the Western Ghats case study. Quantification of non-vertical rates of landscape evolution - and the potential methodologies and proof of concept to implement them – should be of broad interest across the landscape evolution modeling community, and I think the authors have provided a useful contribution advancing towards this goal.

My comments are therefore relatively minor - though I wonder a bit about (1) the necessity to discuss the directional independence of cosmogenic nuclide production, since ultimately the authors simply convert vertical erosion rates to horizontal retreat rates (and this does not depend on any of the systematics of cosmogenic nuclide exposure dating nor complications presented by considering non-vertical production pathways…which, as you will see by my line comments, I think may confuse/over-complicate more than clarify) and (2) methodological details of the river profile analyses (particularly delineation of the escarpment vs. plateau channel reaches and omission of spatially variable rainfall) and retreat rate conversion methods (particularly how local scale products are taken and how erosion of local relief within coastal plain tributary valleys affects the horizontal retreat rate estimates - addressed but not until the end of the discussion section).
**RESPONSE:**
For comment (1), we have greatly simplified Section 2.3.1 by removing the detailed production calculation for non-vertical exhumation pathways. We agree that it is ultimately not needed. We keep the equations of cosmogenic nuclides concentration as expressed with catchment-integrated production and erosion rate by citing published literatures. For comment (2), we reply in the Line-to-Line response below.

**Reviewer:**
- While reading, I had many questions about remnant topography and how erosion of local relief within coastal plain tributary valleys affects the horizontal retreat rate estimates. The authors do not discuss this until line 649, though I feel earlier mention would clarify many points — so I suggest moving this section up, perhaps even into Section 2.3.

**RESPONSE:**
We agree that the local relief discussion comes late in the paper. But moving it up to Section 2.3 would break the flow of the paper for a secondary point, similar to the non-vertical travel paths – although it is important to discuss remnant topography, it affects only interpretation, not methods or calculations. Therefore, we would like to keep it in Section 4 with other discussion points.

**Reviewer:**
- For the Local Scale Product method, what sized elemental surfaces are used and how sensitive are the results to this? Also, how are the normal vectors determined (I imagine by making a slope measurement - over what scale and by what method…steepest descent?)

**RESPONSE:**
In the Local Scalar Product method, we discretize the basin surface into triangular elemental surfaces. The three vertices of a triangular surface are $\{x_i, y_i, z_i\}_{i=1,2,3}$ which define three vectors in space. The normal of the elemental surface is the vector that is normal to the three vectors. We clarified this in Line 595 to line 600 on page 18.

We did not conduct a sensitivity test per se, but do not expect a large sensitivity given that results depend only on how well the surface area is represented, not the actual, local slope. The comparison between the two methods is a type of sensitivity test.

**Reviewer:**
Throughout: I've noticed some inconsistency in the use/absence of oxford commas. Suggest standardizing.

**RESPONSE:**
We have standardized the grammar and spellings.

**Reviewer:**
Line 30 **"their" → "its"**

**RESPONSE:**
Has been corrected.

**Reviewer:**
Line 33: I find the sentence "Relatively static escarpments are supported by low denudation rates." to be a bit confusing and presume it should read something more like "This hypothesized evolution towards relatively static escarpments is supported by the observation of slow time-averaged denudation rates."

**RESPONSE:**
Yes, that's what we meant. We have reworded this sentence accordingly.

**Reviewer:**
Line 38: **"gynomorphically" → "geomorphically"**

**RESPONSE:**
Has been corrected.

**Reviewer:**
Line 52: define DCN (Detrital cosmogenic nuclide?)

**RESPONSE:**
Yes, the DCN is short for detrital cosmogenic nuclides. We specified it since it is the first place it appears.

**Reviewer:**
Line 53: delete comma. Perhaps it'd be helpful in this comparison of rates to use consistent units (e.g. change this and/or rates above to all be reported in m/Ma, km/Ma, or mm/ka).

**RESPONSE:**
We used a constant unit of m/Ma for all rates.

**Reviewer:**
Line 56-57: "Given that the erosion associated with the relief of even the largest escarpments is under 2 kilometers…" confused me. Perhaps "Given that rocks are exhumed from depths of less than 2 kilometers due to retreat of even the largest escarpments, their cooling is likely.."
**RESPONSE:**
We have reworded this sentence.

**Reviewer:**
Line 82: Does "5 to 20 kms of extent normal to the margin" just mean a 5-20 km-wide coastal plain? Perhaps rephrase if so.
**RESPONSE:**
We meant the steep part of the escarpment. We reworded this sentence.

**Reviewer:**
Line 124: I think "on top of the escarpment" could be clarified since I read "on top of the escarpment" to mean "on the plateau." Perhaps "have a headwater divide defined by the escarpment edge" (if I understand correctly?) or maybe " have a headwater divide on the escarpment" (if this is what you mean?)?
**RESPONSE:**
We meant escarpment-draining rivers that have a headwater divide at the escarpment edge, in contrast with escarpment-draining rivers that their head water divide is on the plateau, showing an offset in space between the escarpment edge and the water divide. We clarified this in the revision.

**Reviewer:**
Line 156-157: How sensitive are your results to the definition of this search area? I wonder in particular about the upper bound slope threshold, since there is not a clearly defined break in the data.
**RESPONSE:**
The same search window is used for each basin in our study. This window ensures that it contains the "vague" break between escarpment reach and coastal reaches. Of course, not all basins display clear break on the slope-area plot. If the break is unclear, then the data trend above and below the break should be similar. In this case, the slope regressed from only above the break or include more data below the break will be similar as well. We tested for a few different windows, the results are robust.

**Reviewer:**
Line 159: From a slope-area plot or by collapsing the tributaries onto similar trends in chi-elevation space? Clarify. Guessing the former, but did you also compare your results to the latter?
**RESPONSE:**
They are from slope-area plot. We have clarified this in the revision. We didn't compare best chi-collapse concavities with these slope-area concavities.

**Reviewer:**
Line 166: Perhaps the l in Sl should be subscripted to avoid confusion with e.g. channel length?
**RESPONSE:**

We replaced it with $S_{river}$.

**Reviewer:**
Line 173-174: Suggest clarifying "but not for individual rivers." Does this imply "gradually retreating rivers"?
**RESPONSE:**
It meant individual escarpment-draining rivers. We clarified it in the revision.

**Reviewer:**
Line 176: Pluralize "red square".
**RESPONSE:**
Has been corrected.

**Reviewer:**
Line 180: Should this say "a concavity of 0.42"? Otherwise the main text and figure caption seem to disagree? Or I guess this is the local best-fit concavity rather than the global mean you use in other analyses? Perhaps this should be stated in the figure caption or text to clarify.
**RESPONSE:**
In this original figure, we used concavity of .38 for $\chi$ calculation of this basin. The concavity 0.38 is from the maximum $R^2$ regression of slope-area as shown in Fig. 4c. But to stay consistent with the text and avoid confusion, we recalculated $\chi$ with concavity 0.42. The shape pattern of $\chi$ profiles still remains when a concavity of 0.42 is used. We replaced Fig. 4b with $\chi$ profiles (concavity 0.42) in the revision.

**Reviewer:**
line 196: "the response of temporal" → "the result of temporal" or "they are not a response to temporal". Also suggest adding "they" after "i.e."
**RESPONSE:**
Has been rephrased.

**Reviewer:**
Line 201-205: Here I assume you use a concavity of 0.45 (to give ksn in m^0.9) and not 0.42…? This should be clarified.
**RESPONSE:**
We used a concavity of 0.45 here in order to compare with the global compilation by Kirby and Whipple (2012). We clarified this in the captions of Figure 5.

**Reviewer:**
Line 203: "erosion resistant" → "resistant to erosion"
**RESPONSE:**
Has been rephrased.

**Reviewer:**
Line 205: How do spatial variations in rainfall (i.e. I presume significant orographic rainfall gradient?) potentially affect river profile morphology? Have you tried incorporating rainfall into your chi-elevation analyses?
**RESPONSE:**

Yes, there is significant rainfall gradient with high rainfall on the escarpment side and becoming relatively arid on the plateau side (Mandal et al., 2015). We tried to incorporate rainfall into the $\chi$ calculation. (1) The shape of $\chi$-elevation profiles keeps the same pattern as shown on Figure 3 for both kinds of rivers (the black and the red on Figure 3) after correction for rainfall in $\chi$ calculation. (2) Steepness of escarpment reaches calculated from $\chi$ (with correction of rainfall) is even higher than which is calculated from our slope-area analysis for the same concavity of 0.45. If we plot the steepness (from $\chi$) on Fig. 5, the Western Ghats data cloud will shift upwards, still being among the highest reported steepnesses.

**Reviewer:**
Line 208: Not sure what ESurf convention is but I recall a British "focussed" earlier but an American "focused" here. Suggest standardizing.
**RESPONSE:**
We standardized the spellings to a common convention.

**Reviewer:**
Line 217: What is meant by "conventional"?
**RESPONSE:**
We want to emphasize here that erosion rate is a metric that is conventionally understood as referring to the surface change in vertical direction. Perhaps not very necessary to use the word "conventional" here so we removed it in the revision.

**Reviewer:**
Line 220: Suggest changing "steady" which seems like an odd way to describe the concentration profile. Perhaps "time-invariant" or "…depth does not change over time".
**RESPONSE:**
We changed this to "time-invariant ", although that is the definition of steady, this may not be apparent in this context.

**Reviewer:**
Line 220-221: Suggest rewording "require only a weaker assumption" to perhaps "relax this assumption and require only that equilibrium…".
**RESPONSE:**
We rephrased it.

**Reviewer:**
Line 241: Minor point, but perhaps this should just say cosmogenic nuclides instead of cosmogenic radionuclides since the theory could be applied to stable isotopes (e.g. Helium-3) as well. Perhaps change throughout.
**RESPONSE:**
We changed it throughout the paper.

**Reviewer:**
Line 242-245: This sentence is super long and seems unnecessarily redundant…suggest streamlining and shortening.
**RESPONSE:**
Indeed! We changed it and reorganized Section 2.3.1.

**Reviewer:**

Line 247-248: I think the sentence "A single exponential represents production of nuclide by spallation." needs to be clarified (or I guess I'm confused about whether or not this was already stated above at line 244? I see the need to clarify between production by spallation vs capture of muons for your next point, but I think the sentence should still refer back to eq 3).

**RESPONSE:**

This sentence still refers to Equation (3). In the simplified Section 2.3.1, this expression has been removed.

**Reviewer:**

Line 250-251: Suggest rephrasing "The penetration distance is in a general direction" which I presume just means attenuation of particle rays into a bedrock surface can occur in any direction?

**RESPONSE:**

The particle rays attenuate into a bedrock surface occur in any direction, each direction has its own attenuation length (Lal 1991). In practice, this attenuation process can be modeled as a collimated ray penetrating the target rock and the variable attenuation length is scaled into one effectively representative value to account for the overall production of cosmogenic nuclides (Lal 1991; Dunne et al., 1999). The penetration is scaled to be vertical in the many papers. What we meant was the if one considers a rock parcel exhumed from underground to the surface, the accumulation of nuclides in the rock parcel has to be integrated along the that exhumation path, which is not necessarily vertical. After the simplification of Section 2.3.1, catchment-wide accumulation of nuclides for non-vertical exhumation path is beyond our goal of this study so we have removed this expression in the revision.

**Reviewer:**

Line 265: In the case of a radionuclide, shouldn't eq 7 be $C\_0 = P\_0/(lambda +rho*e/Lambda)$ where lowercase lambda is the decay constant? If this only applies to stable isotopes this should be stated more clearly here (as perhaps it is in line 277?)

**RESPONSE:**

We agree that we simplified this expression by removing the decay constant without carefully pointed out the proper conditions. In the revision, we added the decay factor to Equation (3) on page 11 and clarified that decay factor should only be considered when the cosmogenic nuclide is radioactive on Line 398-399 on page 12. We also pointed out threshold erosion rate that radioactive decay can be ignorable for detrital $^{10}$Be on Line 412-414 on page 12.

**Reviewer:**

Line 277: What is meant by the calculation of production rates? E.g. altitudinal and latitudinal variations? Perhaps you can be more specific here (as I see you are in the following sentences)…

**RESPONSE:**

We have removed this sentence in the simplified Section 2.3.1.

**Reviewer:**

Line 288-289: Huh? I don't understand why local shielding would sum to zero during integration…? This seems to contradict the results of DiBiase 2018, which I now see could be stated more clearly in the preceding sentence (the effect of increasing vertical attenuation

length as a function of hillslope angle and skyline shielding offset the effect of decreasing surface production rate, such that both can effectively be neglected in the calculation of mean production rates for catchments with surface slopes < 30).
**RESPONSE:**
We combine the response to this comment with Line comment 355-370.

**Reviewer:**
Line 343: This is impingement upon the slope/aspect of the surface exposure surface not significantly affecting the vertical attenuation length and/or topographic shielding no? As a (I'm not sure related or unrelated?) example: exposing the base of a cliff whose height exceeds the attenuation depth could expose deeper rock to cosmogenic nuclide accumulation that could not be analogously considered as a product of spatially variable surface lowering…right? Perhaps I'm missing something. This seems like an important consideration and how this plays in (or under what conditions this is applicable) should be stated more explicitly here I think (as is displayed by the theta<30 deg label in Fig 7 and discussed further later).
**RESPONSE:**
Not sure we understand the point exactly, but the shielding issue is no different in a horizontally – retreating catchment as it is in a vertically eroding catchment. Locally, there will be conditions that apparently violate the assumptions, but provided that these variations average out over the catchment, they will not be important. So long as the spatially averaged production is in balance with the erosional flux, results will not vary significantly.
According to DiBiase (2018), skyline shielding impact is significant if valley facets are steeper than 30 degree. Topographic gradient of escarpment basins is smaller than 30 degree in our study area which means that skyline shielding of cosmic rays by the high plateau should have limited impact on the production of nuclides in the escarpment basins. Therefore, we think shielding impacts are negligible for our retreat problems. So we made no shielding corrections in our calculations.

**Reviewer:**
Lines 355-370: I appreciate the acknowledgement of the limitations here, but I still struggle to understand how production calculations can in practice be even quasi-independent of rock motion or the path of integration when one considers the effects of topographic shielding and attenuation length. Perhaps the finite range/variability of e.g. valley spacings or river network morphology limit this to some extent, but I can imagine radically different catchments that could project to the same horizontal (profile) area but different vertical (planform) area or vice versa…and this would seemingly complicate the coordinate transformation and application of equations 8-13 in potentially drastic ways.
**RESPONSE:**
This concept takes some thinking to get around rather non-intuitive concepts. It is true that at any single point, the production and surface concentration will vary from point to point depending on pathway. But the assumption is that these all cancel once the integration is done over the full catchment. The point of this paragraph is to try to make the point that the within-catchment variance of all geometric factors is large. Rock cannot possibly be moving in a constant direction relative to the Earth' surface throughout the catchment. This is the point of Figure 7. We as a community now accept that this variance averages out so that detrital CN gives a meaningful and robust result. The assumptions we are making in this paper are no different. The averaging out simply results in horizontal motion of the surface as a whole, not point by point.

**Reviewer:**

Line 404-412: I'm a bit confused by these statement since any erosion that e.g. occurs within local depressions/ mid-elevation tributary valleys and/or that changes surface concavity but not relief may not be captured in this projection no? Perhaps this should be acknowledged (and/or the limited cases in which this can effectively represent Vrock should be stated) if I understand correctly?

**RESPONSE:**

We think this is referring to the problem of double or triple overlap of projected points. It is correct that mid-slope relief such as small scale interfluves will not change the projected area of the basin as they appear in the projected "shadow" of the main escarpment. Our assumption is that these are simply part of the variance in erosion rate that averages out. This is discussed along with the other occurrences of remnant topography in Section 4.2.1.

**Reviewer:**

Line 424: Delete comma after "catchment surface"

**RESPONSE:**

Has been deleted.

**Reviewer:**

Line 447: Suggest clarifying "an example of the conversion of vertical basin-averaged erosion rates to horizontal escarpment retreat rates".

**RESPONSE:**

Clarified, a little differently.

**Reviewer:**

Line 484: I find it hard to assess the comparability of the results from the two methods in tabulated form. Could you also include a 1:1 plot of the retreat rates for the two methods plots against one another?

**RESPONSE:**

We have made a new figure to show the comparison of retreat rates from the two methods. The figure is now Figure 10 on page 21.

**Reviewer:**

Line 519: Delete first "expected"

**RESPONSE:**

Corrected.

**Reviewer:**

Line 546: Average catchment channel steepness? Or some more local steepness measurement? I see in the figure caption that channel steepness is calculated from regression of slope-area plots using a concavity of 0.42. I assume this is limited to the topography on the coastal plain, as shown by the red points in figure 4? I'd suggest clarifying these details in the text (e.g., "steepness" → "basin-averaged channel steepness on the coastal plain")

**RESPONSE:**

Figure 13 plots the steepness of escarpment-flowing reaches. The steepness is from regression of slope-area plots using a uniform concavity of 0.42. We clarify this by referring to "channel steepness of escarpment reaches" to replace the old "channel steepness".

**Reviewer:**

Line 560: Do the "short solid lines" refer to the error bars? I'd suggest calling them the latter if so to avoid confusion with the dashed line.
**RESPONSE:**
Yes, it referred to the error bars. We have changed it.

**Reviewer:**
Line 562: Units of K should be meters^(1-0.84*n)/yr no?
**RESPONSE:**
We made a mistake here. It has been corrected.

**Reviewer:**
Line 574: "referential" → "reference"
**RESPONSE:**
Has been replaced.

**Reviewer:**
Line 581: Is correct to say "is accounted for in the horizontal retreat calculation"? Perhaps I am missing something (?), but it's not clear to me how this is actually accounted for in these calculations [or, if not, how it affects the inferred retreat rates differently than the flexural uplift of the coastal plain (besides the greater ability of the downstream coastal plain portions of the basins to respond to/erode any flexurally uplifted topography…and perhaps this is the point you are making?)]. In any case, I'd suggest clarifying since I'm otherwise confused about why uplift of the escarpment plateau could not also potentially contribute to the mass balance of the basin for basins whose headwaters do drain the plateau…?.
**RESPONSE:**
Any uplift of the plateau (assuming no erosion of the plateau) is manifested as an increased height of the edge of the plateau. Our retreat rate is based on this height, so the flexural uplift of the plateau is implicit to the retreat calculation, and therefore accounted for. We have reworded this paragraph to explain this more completely on line 810-817 .

**Reviewer:**
Line 610-625: I think it may aid understanding to change the terminology used to describe this calculation. For instance, I'd suggest changing "area" to "volume per unit width of the escarpment" to avoid confusion with planview/drainage area and perhaps also rename the "area ratio" to something more descriptive, perhaps "relative volume per width isostatic". I think this terminology would be particularly useful to clarify the sentences at line 623-625, "Area ratio…" , which confused me (Perhaps instead: "Ratio showing the volume per unit width of the escarpment uplifted by flexural isostasy relative to the volume per width eroded by escarpment retreat for different effective elastic thicknesses. Higher ratios signify greater proportions of un-accounted for isostatic uplift and correspondingly greater overestimation of retreat rates." Does the overestimation of retreat rates indeed scale linearly with the underestimation (omission) of relative isostatic volume per width uplifted (→eroded)? (in other words, does neglecting 10% of the volume per width eroded by ignoring isostasy truly translate to a 10% error in the retreat rates, as seemingly implied by the wording now? I'd guess not…?)
**RESPONSE:**
We appreciate that referring to area might be confusing to readers, so we recast the analysis as mass flux per unit time and per unit width of the escarpment. We carry the unit width and time and density through new Equations 16 -19 on line 850-860. Once converted to a ratio, these terms cancel, but we now refer to this as a mass flux ratio, not an area ratio. Not sure we follow

the final comment, but the quantitative impact is shown in the updated Figure 15 and does scale approximately linearly, which does not mean they are equal.

**Reviewer:**
Lines 642-648: Isn't this also a problem in the Basin Projection Method, which will be insensitive to any local relief (below the scale of the total projected relief of the basin)? I guess the effect here may be of different sign, where neglecting this interior basin erosion via Basin Projection will tend to overestimate horizontal retreat rates rather than produce negative fluxes as in the Local Scalar method case. Perhaps I'm missing something (?), but if not, it seems incomplete to discuss the limitations solely of the Local Scalar method in this regard.
**RESPONSE:**
The Basin Projection Method is insensitive to the local relief that is in the shadow of the total relief in the projection direction. We discussed this in Section 4.2.1

**Reviewer:**
Line 649: Ah ok here's the discussion of many of the points I've been raising regarding internal basin relief. Perhaps this section should be moved up to clarify these points sooner…? I'd suggest moving it up to Section 2.3 even.
**RESPONSE:**
Yes, we agree these are very important points, but they are implications of the analysis and don't impact the methods, so to put them earlier would disrupt the logical development of the paper. We would prefer to keep these in Section 4 Discussions. Hopefully that will keep readers engaged, waiting to have their questions answered!

**Reviewer:**
Line 674: What does a "well-defined regional…form" mean? Suggest clarifying or deleting.
**RESPONSE:**
We meant a basin whose local slopes dip at similar azimuth. We clarified it in the revision.

**Reviewer:**
Line 687: "over a thousand-year timescale" → "over thousand-year timescales"
**RESPONSE:**
Replaced.

**Reviewer:**
Line 689: delete comma
**RESPONSE:**
Corrected.

**Reviewer:**
Line 692-694: "Average retreat rates are in the same range of magnitude as of the DCN 10Be-inferred retreat rates, although they are systematically higher,…" → "Average retreat rates are in the same range, but systematically higher than DCN 10Be-inferred retreat rates, …" Also suggest changing "historical average" to "long-term" or "average since rifting"
**RESPONSE:**
Replaced accordingly.

**Reviewer:**

Line 704: "historical" → "long-term"
**RESPONSE:**
Replaced.

**Reviewer:**
Line 705: "open" → "unknown" (if I understand correctly?)
**RESPONSE:**
Yes, we meant it's still unclear. It has been replaced.

**Reviewer:**
Line 715: "such as occur" → "such as those that occur"
**RESPONSE:**
Corrected accordingly.

Response to Greg Balco

**Greg Balco:**
This paper is quite simple in principle: it points out that a cosmogenic-nuclide measurement in fluvial sediment leaving a basin, which is usually interpreted as a mean erosion rate in the basin, could also be interpreted as a horizontal retreat rate of an basin fronting a retreating escarpment. The cosmogenic-nuclide measurement is telling you is the mass flux out of the basin, and the mass flux could equivalently be the result of either vertical "erosion" or horizontal "retreat." Of course this is correct, and retreat makes more sense than erosion if you are trying to figure out what is happening to an escarpment. The main complication is only that relating a mass flux out of a basin to a horizontal "retreat" of the basin is geometrically fairly complicated, and much of the paper is devoted to unscrambling this issue.

Overall, I think this paper is good and I'm supportive of publishing it. I only found one major issue in review, as follows. Basically, relating a cosmogenic-nuclide concentration to an erosion or retreat rate has two halves: (i) figuring out the cosmogenic-nuclide production rate in the basin, and (ii) parameterizing the mass flux out of the basin. This paper is all about (ii), and simply adopts (i) from previous literature. However, there is a sizeable section of the paper (section 2.3.1, starting line 240) devoted to explaining (i). This is a bit of a problem in the paper, because it contains some vague and confusing elements that make this explanation more confusing than it is in the standard literature describing basin-scale production rate calculations. Specifically, these elements are confusing:

**RESPONSE to the general comments:**
We agree with the comments that several statements were confusing and inaccurate. We have simplified Section 2.3.1. In the simplified Section 2.3.1, we removed the detailed production calculation for non-vertical exhumation pathways. We agree that it is ultimately not needed. We keep the equations of cosmogenic nuclide concentration as expressed with catchment-integrated production and erosion rate by citing published literatures.

**Greg Balco:**
-- Line 249, 'generally taken to be exponential.' In fact this isn't true, because the fact that muon energy increases with increasing depth means that the exponential attenuation length continually increases with depth. You can't approximate this accurately with a finite sum of exponentials. What you can do in integrating production for this application is assume that there is a single exponential function that, when integrated, gives the right answer...but the parameters of that exponential function vary with the erosion rate, so this approach is implicit. I don't think it's necessary to get into this level of detail, but the point is that this statement is oversimplified and confusing.

**RESPONSE:**
We agree that the expression "generally taken to be exponential" was overly simplified and causes more confusion than clarification. We removed this expression in the revision.

**Greg Balco:**
-- Line 250. I don't understand the sentence " The penetration distance is in a general direction...." What does this mean?

**RESPONSE:**

The particle rays attenuate into a bedrock surface propagate in any direction, each direction has its own attenuation length (Lal 1991). In practice, this attenuation process can be modeled as a collimated ray penetrating the target rock and the variable attenuation length is scaled into one effectively representative value to account for the overall production of cosmogenic nuclides (Lal 1991; Dunne et al., 1999). The penetration is scaled to be vertical in the many papers. What we meant was the if one considers a rock parcel exhumed from underground to the surface, the accumulation of nuclides in the rock parcel has to be integrated along that exhumation path, which is not necessarily vertical. After the simplification of Section 2.3.1, catchment-wide accumulation of nuclides for non-vertical exhumation path is beyond our goal of this study so we have removed this expression in the revision.

**Greg Balco:**

-- Line 288-89. "...most shielding is local...and therefore sums to zero during integration.' This statement implies that a shielding factor less than one at some location within the watershed will be balanced by other locations where the shielding factor is greater than one. Shielding factors greater than one are impossible by definition, so this statement makes no sense. This may have something to do with the fallacy that the total cosmic-ray flux impinging on a basin is the same as the flux passing through a horizontal plane at the top of the basin. This isn't true.

**RESPONSE:**

We are not sure what the final statement is referring to as we didn't make any reference to a horizontal surface. The principle that we were trying to convey is that once a particle enters the convex-hull of a drainage basin as define by its perimeter, it either impacts air or the rock, and any local topography within the convex hull will not change this quantity, apart from the change in air/rock ratio. In any case, if the reviewers are satisfied with our assumption that a shielding correction is not needed, we will omit controversial justifications and simply state our assumption.

**Greg Balco:**

-- Line 291-92. It's fine to simplify the math by ignoring radioactive decay, but if you do this you need to be specific about when this assumption is inaccurate. Specifically, this assumption is OK when the quantity E/L (erosion rate in g/cm2/yr divided by attenuation constant in g/cm2) is a lot bigger than the nuclide decay constant lambda (1/yr). I would choose a typical nuclide (Be-10) and specifically indicate for what range of erosion rates this is true. Note this also needs to be dealt with later at line 369.

**RESPONSE:**

We agree that we simplified this expression by removing the decay constant without carefully pointed out the proper conditions. In the revision, we add the decay factor to Equation (2) and pointed out threshold erosion rate that radioactive decay can be ignored for detrital $^{10}$Be. We also have dealt with Line 369 accordingly. We note however, that this is not so simple for the case of spatially variable erosion rate as we have with escarpment retreat, as even with an average erosion rate above the validity threshold, some parts of the basin may have an erosion rate much lower. We specify this point on Line 410-414.

**Greg Balco:**

The point of all this is that section 2.3.1 doesn't add anything to previous literature -- it's not intended to, that isn't what this paper is about -- but it contains several confusing sections that make the reader more confused than if they had just referred to existing literature. I strongly recommend greatly simplifying this section to use a simple form of the integral production equation and refer everything to existing literature rather than getting into the details here. Sure, the question of whether or not to consider muon production is important in understanding the accuracy of an erosion or retreat rate inferred from a Be-10 concentration, but it has nothing to do with what this paper is about, which is representing mass flux as retreat instead of erosion. Summary: greatly simplify this section.

Other than that, I found this paper interesting to read and I'm supportive of publishing it.

**RESPONSE:**

We have simplified Section 2.3.1 accordingly.

**Greg Balco:**

Line 384 -- in this section you don't derive expressions for cosmogenic-nuclide production. Rethink this sentence.

**RESPONSE:**

We have modified this sentence accordingly.

**Greg Balco:**

Figures 9 and 10. It would be helpful to label the methods used for each polar plot (basin projection, local scalar product) on the figure itself, so that the reader doesn't have to refer back and forth between the caption and the figure.

**RESPONSE:**

We have labeled the methods for the two figures.